# Leveraging Gauge Freedom for Learning Non-Gradient Population Dynamics of Stochastic Systems

**Jules Berman** [1]   **Tobias Blickhan** [1]   **Benjamin Peherstorfer** [1]

## Abstract

Existing work on population dynamics inference often focuses on flows arising from vector fields that are the gradients of scalar potentials. Among all admissible flows that are compatible with the population dynamics, gradient flows are optimal in a specific sense: they minimize kinetic energy. The selection of fields based on different criteria corresponds to a gauge freedom when determining population dynamics, which we leverage in this work. We propose Non-Gradient Inference Flows (NGIF), an algorithm to infer non-gradient population dynamics using a weak formulation of the continuity equation. This allows us to parameterize general vector fields and choose other selection criteria beyond minimal kinetic energy. We demonstrate on a variety of low- and high-dimensional physics problems that this more general approach improves distributional accuracy over gradient-restricted baselines and better captures non-potential transport.

## 1. Introduction

We aim to learn models of dynamical systems observed through independent samples at each time point $t$: for each $t$ we are given a set of samples drawn from the time-marginal distribution of the states, but we do not observe trajectories or any correspondence of individual samples across time points. The resulting task is to infer population-level dynamics, i.e., a dynamical model whose induced evolution matches the observed sequence of time-marginal laws rather than individual trajectories.

This setting is relevant for stochastic, chaotic, or turbulent systems in physics and engineering, where individual trajectories are intrinsically unpredictable. In such regimes,

matching time marginals opens the door to learning predictive data-driven dynamical and reduced models (Neklyudov et al., 2023; Berman et al., 2024; Blickhan et al., 2025). This setting is also examined in biology, for example in single-cell experiments where destructive measurements provide cross-sectional snapshots along developmental time (Bunne et al., 2022; Lavenant et al., 2024).

**Setup and population dynamics inference**   The setup we consider is that we have samples from laws $\rho_t \in \mathcal{P}(\mathcal{X})$ on $\mathcal{X} \subseteq \mathbb{R}^d$ over time $t \in [0, T]$. We assume that each law admits a density function, which we also denote as $\rho_t$. The goal of population dynamics inference is to learn a time-dependent velocity field $u_t : \mathcal{X} \to \mathbb{R}^d$ whose induced flow transports $\rho_0$ to the marginals $(\rho_t)_{t \in [0,T]}$. The requirement of matching the time marginals can be expressed as asking the pair $(u_t, \rho_t)$ to satisfy the continuity equation

$$\partial_t \rho_t + \nabla \cdot (\rho_t u_t) = 0, \tag{1}$$

in the sense of distributions. Given such a field with sufficient regularity, one can generate representative dynamics by integrating the flow ODE

$$\frac{\mathrm{d}}{\mathrm{d}t} x_t = u_t(x_t), \qquad x_0 \sim \rho_0, \tag{2}$$

which produces random trajectories whose time-$t$ marginal satisfies $x_t \sim \rho_t$ for all times $t$.

**Gauge freedom**   A central difficulty is that the continuity equation (1) provides only an underdetermined constraint on the velocity field $u_t$. The continuity equation only depends on $u_t$ through $\nabla \cdot (\rho_t u_t)$. All dynamics induced by the velocity that do not change this quantity have no effect on the evolution of the time marginals and are therefore not fixed by asking $u_t$ to satisfy the continuity equation. Concretely, if $u_t$ satisfies (1) and $w_t$ is such that $\nabla \cdot (\rho_t w_t) = 0$ in the sense of distributions, then $u_t + w_t$ yields the same evolution of the time marginals $\rho_t$. This underdetermination of $u_t$ is a gauge freedom: the time marginals determine only parts of the flow, leaving $\rho_t$-weighted divergence-free components unconstrained. Under the usual smoothness, boundary, and domain assumptions, one can view this through a $\rho_t$-weighted Hodge-Helmholtz decomposition; see Chorin & Marsden (1993, Page 37). This splits

[1]Courant Institute of Mathematical Sciences, New York University, New York, NY 10012, USA. Correspondence to: Jules Berman <jmb1174@nyu.edu>.

*Proceedings of the 43rd International Conference on Machine Learning*, Seoul, South Korea. PMLR 306, 2026. Copyright 2026 by the author(s).

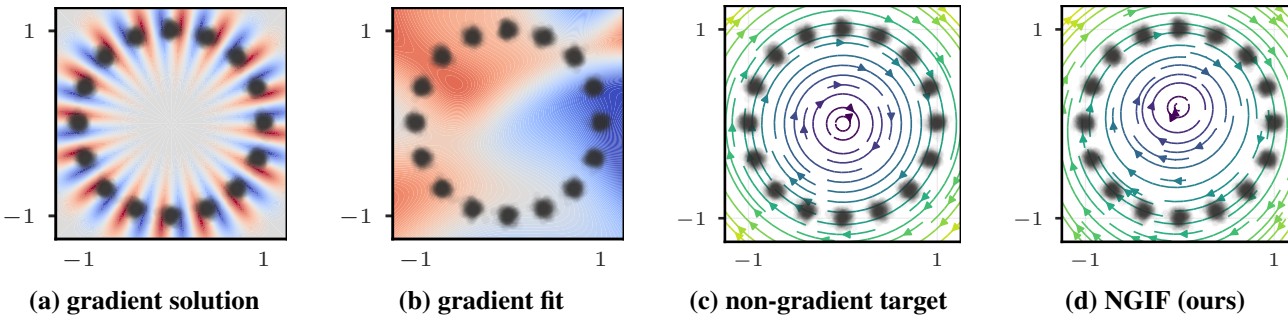

*Figure 1.* We compare various solutions to the Gigli example A, in which a set of Gaussian components arranged on a circle rotates at a constant speed over time. The analytic solution when constrained to using gradient fields in (a) gives a highly irregular target that may be difficult to learn or represent, causing inaccurate approximation of this potential in (b) when we parametrize with a gradient field. In contrast, the unconstrained vector field solution in (c) is easily and accurately approximated with our non-gradient flow in (d). Although the solution in (a) is optimal in terms of kinetic energy, the solution in (c) has only a marginally larger kinetic energy. For a marginal increase in kinetic energy, the non-gradient solution gains much greater regularity.

$L^2(\rho_t; \mathbb{R}^d)$ into the $\rho_t$-weighted divergence-free subspace and its orthogonal complement; the continuity equation determines only the component orthogonal to the $\rho_t$-weighted divergence-free subspace.

**Fixing the gauge with minimal kinetic energy** One canonical choice to fix the gauge is to select the field that has minimal kinetic energy, i.e. setting the solution to zero on the undetermined subspace:

$$\min_{u:\,(1)} \frac{1}{2} \int_0^T \int_{\mathcal{X}} |u_t(x)|^2 \rho_t(x)\, \mathrm{d}x\, \mathrm{d}t$$

Selecting the minimal kinetic energy field is closely related to optimal transport (OT) and, under regularity assumptions, yields a unique velocity field that is necessarily a gradient field $u_t = \nabla s_t$ for some potential $s_t : \mathbb{R}^d \to \mathbb{R}$; see Ambrosio et al. (2005, Proposition 8.4.5).

Much of the existing methodology for learning population dynamics hard-encodes this gauge by parametrizing a potential $s_t$ and learning $u_t = \nabla s_t$ via $s_t$, as in action matching (Neklyudov et al., 2023; Berman et al., 2024) and the discrete inverse continuity equation method (Blickhan et al., 2025). Methods concerned with unbalanced problems and growth functions (Zhang et al., 2025b;a; Sun et al., 2025) use this parameterization. Approaches that build on the Jordan-Kinderlehrer-Otto framework (Jordan et al., 1998) such as JKONet and related works (Bunne et al., 2022; Alvarez-Melis et al., 2022; Terpin et al., 2024; Persiianov et al., 2025) also restrict the learning to these gradient-type dynamics; see Section 4.

**Going beyond gradient dynamics and minimal-kinetic-energy gauges matters** While the minimal kinetic energy field provides a principled default, it is only one gauge choice. In particular, it can impose an inductive bias that is at odds with the underlying dynamics when rotational components are essential, such as in Hamiltonian systems or incompressible fluid dynamics. This failure case is also mentioned by Terpin et al. (2024, Section G.2).

Because the continuity equation (1) constrains only the compressible part of the flow, selecting the minimal kinetic energy gauge suppresses rotation by construction. In extreme cases, enforcing a gradient structure can turn simple rotational motion into highly oscillatory, pathological potentials whose gradients match the motion only on (or near) the support of $\rho_t$ (Gigli, 2011). Such targets are difficult to represent and learn (Figure 1), and small approximation errors in a highly oscillatory potential can produce large deviations in the generated dynamics.

In addition, hard-encoding $u_t = \nabla s_t$ by parametrizing the potential $s_t$ often complicates optimization because one must backpropagate through spatial derivatives of the network output and because the parameterization restricts the hypothesis class in ways that are not aligned with the data.

The literature on learning non-gradient dynamics is scarce. The recent work Petrović et al. (2025) targets learning non-gradient flows by introducing a prescribed non-gradient reference flow as a prior in the form of velocity information at the sample level. In contrast, we do not require a reference flow or sample velocities. Instead we expose the gauge degree of freedom directly, allowing us to select from a broad family of gauge regularizers, including but not limited to minimal kinetic energy.

A related approach is to endow a flow-matching model with a non-trivial metric structure to obtain curved sample trajectories. This is a slightly different problem, as these works typically operate in the two-marginal setting. We refer to (Scarvelis & Solomon, 2023) and (Kapusniak et al., 2024, Section 6) for a review.

**Our approach: non-gradient flow inference (NGIF) with explicit gauge regularization** We propose a non-gradient approach to population dynamics inference that matches the time-marginal evolution through a moment-based (weak) form of the continuity equation, without restricting the velocity field to gradient form.

For a set of test functions, we require that the rate of change of their expectations under the time marginals agrees with what the candidate velocity field predicts via transport. This yields an objective that can be estimated empirically from samples of the time marginals alone, using random Fourier feature test functions that lead to an informative and computationally cheap empirical loss.

The key feature of our approach is that the weak continuity equation loss separates time-marginal matching from velocity field selection. By construction, our loss only constrains the part of the field that affects the time marginals, leaving the gauge freedom corresponding to divergence-free components untouched. Rather than resolving this non-uniqueness implicitly by restricting the parametrization (e.g., parametrizing the potential instead of the velocity field), we use explicit gauge regularizers. Importantly, the regularizer is a free design choice to support informative inductive bias beyond minimal kinetic energy such as encouraging smoothness and controlling divergence or curl. This enables learning non-gradient fields that can better reflect dynamics with substantial rotational structure, avoiding the pathological behavior that can arise when rotation is mimicked by rapidly changing time-dependent gradient fields. To our knowledge, our approach is the only population dynamics inference method that allows one to explicitly leverage this gauge freedom.

**Contributions**

(a) We separate matching time marginals from gauge fixing by formulating population dynamics inference through weak continuity equation constraints and addressing non-uniqueness with an explicit, freely chosen gauge regularizer.

(b) We instantiate the weak constraints with random Fourier feature test functions, yielding a scalable loss.

(c) We enable learning non-gradient velocity fields that improve representability and stability and, compared to gradient-only formulations, lead to more accurate population dynamics inference.

We release a sample implementation of our method here: `https://github.com/julesberman/ngif`

## 2. Inferring non-gradient fields

Motivated by the need to allow non-gradient dynamics, we propose to learn a velocity field $u_t$ via the weak form of the continuity equation, which allows us to leverage the gauge freedom to impose structural biases other than gradient dynamics on the field that we want to learn.

### 2.1. Weak continuity equation as a moment equation

We seek a velocity field $u_t$ such that the time marginals $\rho_t$ satisfy the continuity equation (1) in the sense of distributions. Let $\phi$ be a test function taken from a test space $\Phi$ so that the weak form of (1) is

$$\int \phi(x)\left(\partial_t \rho_t(x) + \nabla \cdot (\rho_t(x)u_t(x))\right) \mathrm{d}x = 0,$$

for all $t \in [0, T]$. Assuming standard regularity to justify differentiation under the integral, we pull out the time derivative in the first part and integrate by parts in the second term to obtain $\int \phi \nabla \cdot (\rho_t u_t) = -\int \nabla \phi \cdot u_t \rho_t$ and

$$\frac{\mathrm{d}}{\mathrm{d}t} \int \phi(x)\rho_t(x)\mathrm{d}x = \int \nabla \phi(x) \cdot u_t(x)\rho_t(x)\mathrm{d}x. \quad (3)$$

If $\mathcal{X} = \mathbb{T}^d$, then there is no boundary and the boundary term vanishes identically for all smooth periodic $\phi \in C^\infty(\mathbb{T}^d)$. If $\mathcal{X} = \mathbb{R}^d$, then the boundary term vanishes under integrability and tail assumptions, e.g., that $\rho_t u_t \in L^1(\mathbb{R}^d)$ with test functions $\phi \in C_b^1(\mathbb{R}^d)$. Recall that $C_b^1(\mathbb{R}^d) = \{\phi \in C^1(\mathbb{R}^d) \,|\, \|\phi\|_\infty < \infty, \|\nabla \phi\|_\infty < \infty\}$ is the space of all continuously differentiable functions on $\mathbb{R}^d$ with bounded values and bounded first derivative. The condition $\rho_t u_t \in L^1(\mathbb{R}^d)$ implies finite momentum in the system, which is often a natural assumption.

Under these conditions, we can write (3) as

$$\frac{\mathrm{d}}{\mathrm{d}t}\mathbb{E}_{x \sim \rho_t}[\phi(x)] = \mathbb{E}_{x \sim \rho_t}[\nabla \phi(x) \cdot u_t(x)], \quad (4)$$

for test functions $\phi \in \Phi$. The quantity $\mathbb{E}_{\rho_t}[\phi(x)]$ is a generalized moment of $\rho_t$ in the sense that it is the integral of $\rho_t$ against a test function $\phi$. Ordinary moments are recovered as the special case that the test functions are monomials, e.g., $\phi(x) = x$ gives the mean, and $\phi(x) = xx^\top$ gives the second moment. We allow a richer set of test functions $\Phi$ in the following. Furthermore, the moment equation (4) does not just constrain a static expectation at a single time but specifies the time derivative of each generalized moment in terms of $\rho_t$ and $u_t$.

### 2.2. Minimizing weak-form residual

We now operationalize the weak moment identity by enforcing it in expectation over test functions with features sampled from a feature distribution $\nu$.

Let $\Phi$ denote an admissible test space and let $\nu$ be a probability measure so that $\phi_\omega \in \Phi$ with $\omega \sim \nu$ (see forthcoming Section 2.4). We introduce a residual

$$R_t(\phi; v) = \frac{\mathrm{d}}{\mathrm{d}t} \mathbb{E}_{x \sim \rho_t}[\phi] - \mathbb{E}_{x \sim \rho_t}[\nabla\phi(x) \cdot v_t(x)].$$

In the ideal case where $(\rho_t, u_t)$ satisfies the continuity equation in weak form, we have $R_t(\phi; u) = 0$ for all $\phi \in \Phi$ and $t \in [0, T]$.

We can also build on the Fokker-Planck equation, in which case the residual has an additional diffusion term. The parameter $\varepsilon$ is an optional hyperparameter; see Appendix F.1 for details.

We can turn this into a loss by minimizing the residual norm over the random test functions

$$\mathcal{L}(v) = \int_0^T \mathbb{E}_{\omega \sim \nu}\left[|R_t(\phi_\omega; v)|^2\right] \mathrm{d}t. \tag{5}$$

Minimizing the loss $\mathcal{L}(v)$ drives the residual $R_t$ to be orthogonal to the test space. Whether the existence of a minimizer with $\mathcal{L}(u) = 0$ implies compatibility with the continuity equation depends on the richness of the test space that is covered by features sampled from $\nu$. In particular, we require that the sampled test functions are rich enough to determine the weak form on $\Phi$, i.e., they need to generate a dense subset of $\Phi$, using the fact that $\phi \mapsto R_t(\phi; v)$ is continuous. For the torus, when $\nu$ has full support on $\mathbb{Z}^d$, this is sufficient. For $\mathcal{X} = \mathbb{R}^d$, we find an analogous situation for $\nu$ Gaussian on $\mathbb{R}^d$.

## 2.3. Leveraging gauge freedom

The weak continuity constraints determine the velocity field only up to components that do not affect the time-marginal evolution. As a result, there is freedom to impose additional structural bias. While other methods (see Section 1) often are restricted to minimal kinetic energy as their structural bias, our approach allows full flexibility.

**The continuity equation has gauge freedom** The weak form (and hence the loss $\mathcal{L}$) depends on $u_t$ only through the weighted divergence $\nabla \cdot (\rho_t u_t)$. Equivalently, it only constrains the action of $u_t$ through the flux $\rho_t u_t$ appearing in the continuity equation. Consequently, if $u_t$ is compatible with the weak continuity equation (e.g. achieves $\mathcal{L}(u) = 0$), then so is any perturbed field $u_t + w_t$ such that $\nabla \cdot (\rho_t w_t) = 0$ (in a weak sense), i.e., for any test function $\phi \in \Phi$,

$$\mathbb{E}_{\rho_t}\left[\nabla\phi \cdot (u_t + w_t)\right] = \mathbb{E}_{\rho_t}\left[\nabla\phi \cdot u_t\right],$$

because $\mathbb{E}_{\rho_t}\left[\nabla\phi \cdot w_t\right] = -\langle \phi, \nabla \cdot (\rho_t w_t)\rangle = 0$. Therefore $u_t + w_t$ yields the same weak-form residuals and achieves

the same loss. This non-uniqueness of the loss is induced by the natural gauge freedom of the continuity equation, because the continuity equation specifies $u_t$ only up to $\rho_t$-weighted divergence-free perturbations.

**Leveraging the gauge** Recovering a unique velocity field requires an additional selection principle or gauge fixing. Concretely, we select $u$ as the solution of

$$\min_u \mathcal{L}(u) + \lambda \mathcal{G}(u), \tag{6}$$

where $\mathcal{G}$ varies depending on the selected gauge.

**Uniqueness versus non-uniqueness** For certain $\mathcal{G}$ (strictly convex and coercive with $\lambda > 0$), the objective of (6) determines a unique $u$. However, in general, uniqueness is not required. The role of the gauge regularizer is to impose a structural (inductive) bias on admissible solutions. We will show in our numerical experiments that in many practically relevant settings, we intentionally employ gauge regularizers that are non-coercive and therefore do not yield uniqueness. Even when the optimization problem (6) has multiple minimizers, the regularizer restricts the solution set in a principled way by promoting specific structure. For instance, penalizing divergence $\nabla \cdot u$ typically does not lead to an optimization problem (6) with a unique solution, yet it encourages approximately incompressible behavior and leads to trajectories that align with prior physical knowledge about the system, as shown in our numerical experiments.

**Proposed gauge regularizer** In the following, we consider three gauge regularizers. First, we consider a kinetic energy regularizer,

$$\mathcal{G}_{\mathrm{KIN}}(v) = \frac{1}{2} \int \|v_t\|^2 \rho_t(x) \, \mathrm{d}x \, \mathrm{d}t. \tag{7}$$

Importantly, we use kinetic energy as a soft gauge-fixing term rather than enforcing it by restricting the model to gradient fields only, which as shown in Section A can lead to challenging optimization problems. Second, we can penalize the antisymmetric part of the Jacobian of the velocity field, which measures rotation and non-gradient components of the flow,

$$\mathcal{G}_{\mathrm{CURL}}(v) = \frac{1}{2} \int \left\|\nabla v_t(x) - (\nabla v_t(x))^\top\right\|_F^2 \rho_t(x) \, \mathrm{d}x \, \mathrm{d}t. \tag{8}$$

Using $\mathcal{G}_{\mathrm{CURL}}$ penalizes rotation and biases the solution toward gradient-like dynamics but still avoids an explicit potential parametrization that would hard-code gradient-only dynamics. Third, we consider the divergence regularizer,

$$\mathcal{G}_{\mathrm{DIV}}(v) = \int |\nabla \cdot v_t|^2 \rho_t(x) \, \mathrm{d}x \, \mathrm{d}t, \tag{9}$$

which promotes approximately incompressible velocity fields. It is especially useful in fluid-like transport (e.g., tracer particles in a fluid flow field).

## 2.4. Choosing a test space via random Fourier feature test functions

The key ingredient is to construct a finite number of test functions $\{\phi_i\}_{i=1}^M$ for which the weak continuity equation yields informative low-variance constraints and for which the required derivatives are tractable. We consider Fourier modes $x \mapsto \exp\left(i\omega^\top x\right)$ for $x, \omega \in \mathbb{R}^d$, with real and imaginary parts $\cos(\omega^\top x)$ and $\sin(\omega^\top x)$.

The appropriate choice of frequencies $\omega$ depends on the spatial domain: on a periodic domain $\mathcal{X} = \mathbb{T}^d = [0, 2\pi)^d$ one must use integer frequencies $\omega = k \in \mathbb{Z}^d$ so that $x \mapsto \exp\left(ik^\top x\right)$ (and hence $\sin(k^\top x), \cos(k^\top x)$) is well-defined and $2\pi$-periodic in each coordinate. In this case the Fourier modes belong to $C^\infty(\mathbb{T}^d)$ and form the canonical trigonometric dictionary on the torus. On $\mathcal{X} = \mathbb{R}^d$ one may choose $\omega$ from $\mathbb{R}^d$. These functions for $\omega \in \mathbb{R}^d$ are $C^\infty$. Furthermore, for any fixed $\omega$, $\cos(\omega^\top x)$ and $\sin(\omega^\top x)$ belong to $C_b^1(\mathbb{R}^d)$ with $\|\phi\|_\infty \le 1$ and $\|\nabla\phi\|_\infty \le \|\omega\|$.

**Random Fourier feature test functions** We approximate these test spaces using random features. We use paired real features

$$\sin(\omega^\top x), \qquad \cos(\omega^\top x), \qquad (10)$$

where $\omega \sim \nu$ is sampled from a chosen frequency distribution $\nu$. Pairing $\sin$ and $\cos$ (equivalently, using both real and imaginary parts of $e^{i\omega^\top x}$) yields a lower-variance estimator than single complex features and is standard in random-feature approximations of shift-invariant kernels (Sutherland & Schneider, 2015).

**Multiscale random Fourier feature test functions** To generate random Fourier feature test functions, we sample frequencies across multiple bandwidths to improve sensitivity to structure across multiple resolutions in the data, following Parra & Tobar (2017). To do so, we select bandwidth bounds $\sigma_{\min}$ and $\sigma_{\max}$ and construct $B$ logarithmically spaced bandwidths, $\sigma_b = \sigma_{\min} \left(\sigma_{\max}/\sigma_{\min}\right)^{\frac{b-1}{B-1}}$, $b = 1, \ldots, B$. We then sample an equal number of frequencies from corresponding distributions

$$\omega_{i,b} \sim \mathcal{N}\left(0, \; \sigma_b^{-2} I_d\right) \qquad (11)$$

to get a total of $M/2$ frequencies, which then give the $M$ test functions $\{\phi_r\}_{r=1}^M$ using the paired real features (10). On normalized periodic domains represented as $[-1, 1]^d$, we first sample raw frequencies $\hat{\omega}_i$ as in (11) and then set $\omega_i = \pi \lfloor \hat{\omega}_i/\pi \rceil$ coordinatewise for $i = 1, \ldots, M/2$. This forces each coordinate of each frequency to be a scalar multiple of $\pi$, ensuring the corresponding test functions are periodic.

## 2.5. Empirical loss

Let us now consider the empirical setting, which means we have training data $\{x_{t_k}^{(i)}\}_{i=1}^N$ in the form of samples from $\rho_{t_k}$ for time steps $0 = t_0 < t_1 < \cdots < t_K = T$. We now discuss how the loss (6) together with random Fourier feature test functions leads to an efficient training objective.

**Pre-computing test-function moments and derivatives** Given $M$ test functions $\phi_1, \ldots, \phi_M$ as obtained, e.g., in the previous paragraph, we define the empirical estimator of the moment $\mathbb{E}_{\rho_{t_k}}[\phi]$ as

$$\hat{\mu}_{k,r} = \frac{1}{N} \sum_{i=1}^N \phi_r(x_{t_k}^{(i)}), \qquad (12)$$

for all time steps $k = 0, \ldots, K$.

There are many ways to estimate the time derivative $\frac{\mathrm{d}}{\mathrm{d}t} \mathbb{E}_{\rho_{t_k}}[\phi_r]$. However, we use the empirical estimates (12) for estimating the time derivative, which can be noisy. We thus use a smoothing spline: For each $r$, fit a spline in time to $\{(t_k, \hat{\mu}_{k,r})\}_{k=0}^K$, and denote the fitted function by $\hat{\mu}_r(t)$. The time derivative $\frac{\mathrm{d}}{\mathrm{d}t} \mathbb{E}_{\rho_{t_k}}[\phi_r]$ is then obtained from the spline and denoted as $\dot{\hat{\mu}}_r(t_k)$.

The moment estimates $\hat{\mu}_r$ and their time derivatives $\dot{\hat{\mu}}_r$ do not depend on the velocity field. Thus they can be pre-computed across the entire training dataset. This allows us to use all samples for estimating the moments, rather than mini-batching, which helps to reduce the variance during training iterations.

**Empirical loss** We now parametrize the velocity $u$ as $u_\theta : \mathbb{R}^d \times [0, T] \to \mathbb{R}^d$ with a neural network. Let us consider the empirical loss for time steps $k = 0, \ldots, K$ as

$$L_k(\theta) = \frac{1}{M} \sum_{r=1}^M \ell\left(\dot{\hat{\mu}}_r(t_k), \widehat{\mathbb{E}}_{x \sim \rho_{t_k}}\left[\nabla\phi_r(x)^\top u_\theta(x, t_k)\right]\right).$$

Here $\widehat{\mathbb{E}}_{x \sim \rho_{t_k}}$ denotes the empirical sample average over $\{x_{t_k}^{(i)}\}_{i=1}^N$ or over a minibatch during stochastic training. The function $\ell(x, y)$ denotes a suitable discrepancy measure. We propose to use a dual-relative loss $\ell(x, y) = \|x - y\|^2/(\|x\|^2 + \|y\|^2 + \epsilon_{\text{loss}})$ with a small tolerance $\epsilon_{\text{loss}} > 0$, which is scale-normalized and remains well-behaved when the magnitudes of different moments and their time derivatives vary widely. Such scale disparities arise naturally, for example, when a test function is not aligned well with the data and thus yields near-zero moments or derivatives or when $\|\omega_i\|$ is large, yielding disproportionately large moments or derivatives.

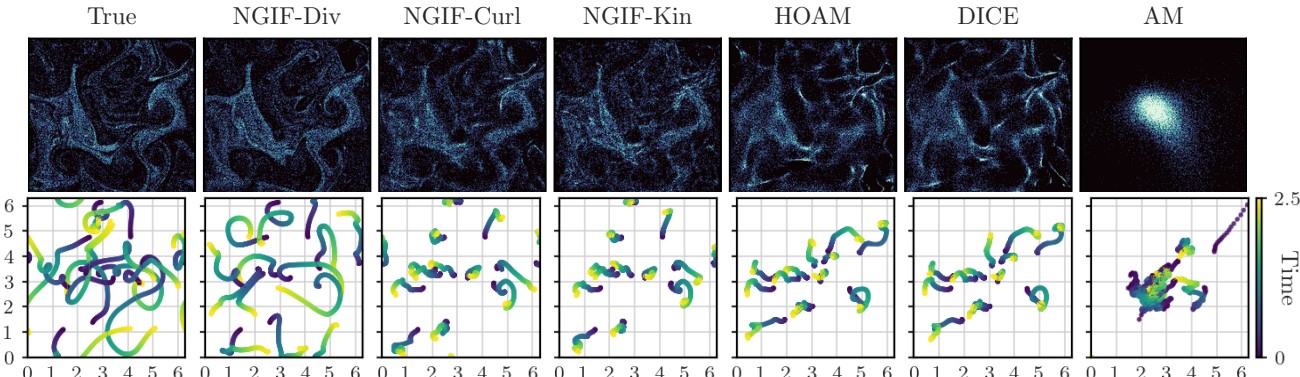

*Figure 2.* Results for the tracer particle case. We see that the divergence regularizer (9) leads to both the most faithful reconstruction of the particle density (top row) and qualitatively similar sample trajectories (bottom row). Note that the physical sample velocity field is divergence-free in this case.

We can then combine the loss functions $L_k$ over time into an empirical loss and add a gauge regularizer to obtain our training problem

$$\min_\theta \frac{1}{K+1} \sum_{k=0}^{K} L_k(\theta) + \lambda \mathcal{G}(u_\theta).$$

**Computational cost of random Fourier test spaces** A key advantage of using random Fourier feature test functions is that all required differential operators for evaluating the loss are available in closed form and are inexpensive to evaluate. In particular, every evaluation of $\phi_i(x)$, $\nabla \phi_i(x)$, or $\Delta \phi_i(x)$ requires only the inner product $\omega_i^\top x$ and simple scalar nonlinearities; the cost therefore scales linearly in the ambient dimension $d$; see Appendix E for details.

## 3. Numerical experiments

### 3.1. Gigli's example

Let us revisit the example shown in Figure 1. It exhibits a family of time marginals of Gaussians with means uniformly on the unit circle. Over time, the Gaussian means rotate along the unit circle, which is an evolution that can be described by a rotational velocity field. However, if one insists on representing the time-marginal flow velocity with a gradient field, then any such gradient representation must feature increasingly sharp gradients as the number of Gaussians grows, even though the underlying dynamics are simple; see Appendix A for details. Figure 1b shows that learning a potential $s_t$ for a gradient field $u_t = \nabla s_t$ fails to recover the highly oscillatory potential that explains this rotation (Figure 1a). In contrast, if we learn a non-gradient velocity field with our approach (Figure 1d) using the CURL regularization, then we achieve a velocity field that is in close agreement with an analytic velocity field that explains rotation (Figure 1c). See Appendix A for details about the learning setup for this example.

### 3.2. Tracer particles in a turbulent field

We consider a time-dependent velocity field $v_t^{(\text{jax-cfd})}$ generated by Kolmogorov forcing as explained in Dresdner et al. (2022); see also Appendix D.1. We generate a training data set $\{x_{t_k}^{(i)}\}_{i=1}^N$ by integrating the ODE $\frac{\mathrm{d}}{\mathrm{d}t} x_t = v_t^{(\text{jax-cfd})}(x_t)$ and collect samples at the discrete times $0 = t_0 < t_1 < \ldots < t_K = T$ with $x_{t_0}^{(i)}$ drawn from a standard Gaussian. The generated samples correspond to tracer particles following this turbulent flow field. In accordance with the assumptions throughout this work, we do not use the trajectory information, instead selecting samples at random across different time marginals.

We derive velocity fields from the data with our approach and gauge regularizers on kinetic energy (7), divergence (9), and curl (8). Additionally, we impose gradient structure directly by parameterizing the potential $s_t$ and using $\nabla s_t$ in the loss (5) instead of parameterizing $u_t$. Figure 2 (top) shows new sample populations generated with the learned velocity fields. Figure 3 shows a measure of the distance between the predicted distribution of particles and the data distribution over time. First, imposing gradient structure by learning the potential leads to poor results. Second, notice with the freedom to use more general gauge regularizers, we can learn different particle dynamics. In particular, the field learned with the divergence regularizer (9) generates particles that behave similarly to the training data particles. This is in agreement with the fact that a low divergence velocity field is a natural structural prior in this case, as we know the generating vector field is divergence-free. Notice that the results obtained with the kinetic and curl regularizer look similar, which is not a coincidence but the corresponding flows agree under certain conditions; see Appendix B.

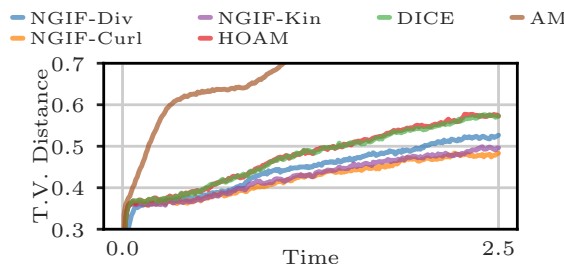

*Figure 3.* Total variation distance of particle histograms over time against ground truth samples. All variants of NGIF outperform the gradient-constrained baselines.

*Table 1.* Relative error in electric energy for the Vlasov-Poisson test cases. Best results are **bold**, second best are *italic*. Top five rows are taken from (Blickhan et al., 2025).

|  | two-stream ↓ | bump-on-tail ↓ | strong Landau ↓ |
|---|---|---|---|
| CFM | 1.44 | 5.52 | 1.96 |
| NCSM | 0.245 | 0.626 | 3.58 |
| AM | 0.275 | 0.892 | NaN |
| HOAM | 0.078 | 0.427 | 0.784 |
| DICE | 0.070 | 0.283 | 0.735 |
| NGIF-Kin (ours) | *0.062* | **0.143** | **0.542** |
| NGIF-Curl (ours) | **0.043** | *0.152* | 0.864 |
| NGIF-Div (ours) | 0.065 | 0.295 | *0.701* |

### 3.3. Vlasov-Poisson instabilities

Let us now consider Vlasov-Poisson systems that evolve interacting charged particles; we follow the setup described by Berman et al. (2024, Section B.2). In particular, we consider the two-stream instability and the bump-on-tail instability in two-dimensional phase space and strong Landau damping in six-dimensional phase space. The problems are parametrized by a characteristic (Debye) length parameter $\mu$, and we evaluate generalization across values of $\mu$.

The fundamental feature of these problems is a transfer of kinetic energy into potential (electric) energy, with the latter growing by several orders of magnitude throughout the simulation. The metric of interest we compute is therefore the evolution of electric energy and we report the relative error in Table 1. Notice the three variants of NGIF all outperform the competing methods, most notably outperforming AM, HOAM and DICE all of which parameterize with the gradient of a scalar potential. In Figure 4 we see again how different regularizers correspond to different learned sample dynamics. The divergence regularizer is able to learn particle trajectories that exhibit richer movement, which helps the learned flow form the final distribution more accurately.

### 3.4. High-dimensional turbulence simulation

We consider the forced turbulent flow setup from Section 3.2 on the flat torus $[0, 2\pi)^2$, integrating to final time

$T = 6.25$. Rather than observing advected particles, we observe cross-sectional snapshots of the vorticity field, a state with dimension $d = 128 \times 128$. We embed these fields into a learned $16 \times 16 \times 1$ latent space via an autoencoder (Appendix G.5) and apply NGIF-Kin to the resulting latent marginals.

Figure 5 visualizes a rollout generated by integrating the learned velocity field starting from an initial latent sample. This rollout should not be interpreted as recovering the true sample trajectory of the turbulent system; in this regime, individual trajectories are dominated by fast, chaotic advection and are not the target of our method. Instead, the rollout is representative in the sense that its law is designed to match the observed time-marginal evolution. To assess this population-level agreement, we track the enstrophy of each frame over time and compare its evolution to that of the true dynamics. The enstrophy curves show that NGIF accurately captures the distributional decay, demonstrating that the learned flow reproduces the salient marginal statistics even though the physical sample paths are not recovered.

This experiment highlights the distinction between sample dynamics and population dynamics in turbulence. The simulation exhibits rapid advection on short time scales alongside a slower evolution of coarse quantities such as enstrophy; population dynamics inference targets the latter.

## 4. Related methods

Population dynamics inference has been studied extensively in computational biology. Early approaches combined neural ODE parameterizations with snapshot matching (Hashimoto et al., 2016) or learned couplings between consecutive marginals via optimal transport (Schiebinger et al., 2019). Closer to our approach are methods that infer an admissible velocity field for a predetermined path $t \mapsto \rho(t)$; see Section 1 (Fixing the gauge with minimal kinetic energy).

*Schrödinger bridges.* The stochastic analogue to JKO methods is known as Schrödinger bridge matching (Chen et al., 2019; 2023; Shen et al., 2025; Hong et al., 2025): successive time marginals are not connected by optimal transport maps but by stochastic processes with endpoints $\rho(t_j)$ and $\rho(t_{j+1})$. The inference step turns into a stochastic differential equation (SDE) integration with the noise level taking the role of a hyperparameter. The vector fields learned by these methods obey a slightly different selection criterion (Léonard, 2014, Definition 2.1). They are also of gradient form. The variant of our method built on the Fokker-Planck equation discussed in Section F.1 is reminiscent of these.

*Relation to moment matching*: More broadly, our approach

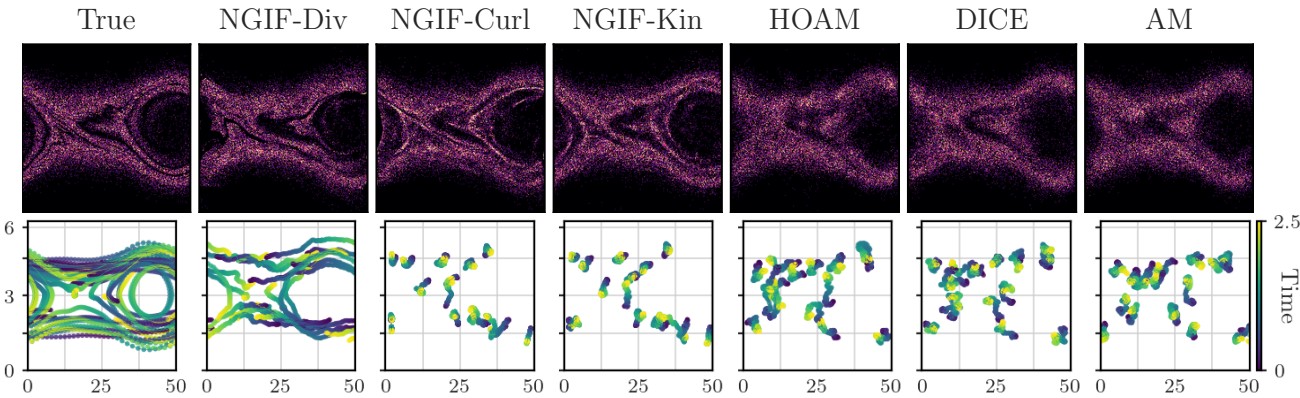

*Figure 4.* Vlasov-Poisson two-stream test case: Our method with any of the regularizers (7)–(9) leads to sharper, more accurate solution fields than methods that parameterize with gradient fields (HOAM, DICE, AM). Note that NGIF-Curl, NGIF-Kin, HOAM, DICE, and AM lead to fields with low kinetic energy with little rotation as expected, while our NGIF-Div does not penalize rotational dynamics.

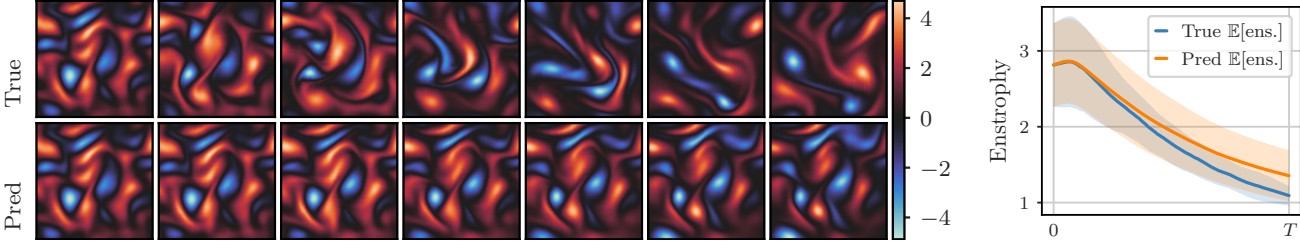

*Figure 5.* High-dimensional turbulence simulation: we plot a true vorticity trajectory next to a rollout generated by integrating the NGIF-Kin velocity field from an initial sample (left). The NGIF rollout is not expected to match the true vorticity trajectory, since the objective is population dynamics inference from time-marginal snapshots rather than trajectory recovery. To evaluate population-level agreement, we compare the enstrophy (right), yielding a 1D distribution at each time point; its mean and standard deviation are plotted, and our NGIF accurately captures the distributional changes in enstrophy.

is related to moment matching; however, instead of matching two distributions (e.g., reference and a target distribution) by regressing on test-function moments, as in kernel mean embeddings and maximum mean discrepancy (MMD) (Gretton et al., 2012) and their random-feature approximations, we match how these generalized moments change over time in a way that is consistent with transport under a velocity field. Thus, our objective is not a static discrepancy aiming to connect a reference and a target distribution. Rather, it is a time-dependent constraint coming from the weak continuity equation. This perspective is complementary to recent moment-matching generative modeling (Zhou et al., 2025), which targets the static setting of sampling from a target, whereas we use moment constraints to infer population dynamics.

## 5. Conclusions and limitations

Population dynamics inference is inherently underdetermined because enforcing the continuity equation to match time marginals constrains only the flux divergence and therefore admits a whole family of velocity fields. Our main contribution is a formulation that separates marginal matching, enforced through weak (moment-based) continuity constraints, from selecting a velocity field with a specific structural inductive bias, implemented via an explicit gauge regularizer. This decoupling makes the inductive bias a transparent modeling choice and avoids hard-wiring the minimal-kinetic-energy/gradient-field gauge, which can be inappropriate when rotational components are essential and can lead to irregular, hard-to-learn targets. Overall, the approach provides a practical route to learn population-level dynamical models from samples of time marginals alone, supporting data-driven and reduced modeling in settings where trajectory data are unavailable or uninformative.

A constrained formulation, minimizing the gauge regularizer subject to the weak continuity constraints, is a natural alternative to our penalized objective (6). In practice, however, hard-constraining neural-network-parameterized velocity fields typically requires additional constrained optimization machinery, such as augmented Lagrangian or inner-outer procedures. The penalized formulation instead gives a single unconstrained objective that can be optimized with standard stochastic-gradient methods, is simple to implement, and empirically enforces marginal matching across our benchmarks. Once marginal matching is han-

dled by the weak continuity-equation loss, the role of the gauge regularizer is to select among admissible trajectory-level dynamics by encoding a prior.

This perspective makes gauge selection a modeling choice rather than a search for a universally correct regularizer. Kinetic energy, curl, and divergence penalties can all yield accurate marginal evolution, but they express different physical preferences; selecting among gauge-equivalent fields with similar distributional accuracy requires trajectory-level knowledge beyond marginal data alone. When such knowledge is available, NGIF can incorporate it directly through the gauge penalty. For example, if the velocity field is known to have a prescribed spatiotemporal divergence profile $g(t, x)$, one can replace the incompressibility penalty by a regularizer of the form $\int |\nabla \cdot u_t(x) - g(t, x)|^2 \rho_t(x) \, \mathrm{d}x \, \mathrm{d}t$, and the resulting validation error can also help select $\lambda$. Our experiments illustrate this explicitly. In the Vlasov example, the divergence gauge permits richer non-gradient particle motion while preserving distributional accuracy. In the tracer experiment, the divergence penalty corresponds to the special case $g \equiv 0$, giving an incompressibility prior and producing trajectories that are qualitatively closest to the physical ones. We view this modular choice of gauge as a key feature of NGIF and as an opening for problem-specific regularizers beyond the three considered here.

*Limitations.* (i) In practice we can enforce the weak continuity equation only against a finite collection of test functions, and it is generally unclear how large this collection must be to yield an informative constraint set for a given problem. While random Fourier tests are inexpensive to evaluate and perform well in our experiments, their informativeness can degrade in very high dimensions, potentially requiring additional prior structure. (ii) Several useful gauge regularizers depend on spatial derivatives of the learned field. Although the weak continuity equation loss avoids differentiating the network, these gauges reintroduce Jacobian computations that can become challenging in very high dimensions. (iii) The trade-off parameter $\lambda$ in (6) must be chosen in practice. A full study of adaptive penalty schedules or constrained variants that solve $\min_u \mathcal{G}(u)$ subject to (4) is an interesting direction for future work.

## Impact Statement

This paper presents work whose goal is to advance the field of Machine Learning. There are many potential societal consequences of our work, none of which we feel must be specifically highlighted here.

## Acknowledgments

The authors have been funded in part by the Air Force Office of Scientific Research (AFOSR), USA, award FA9550-24-1-0327.

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

# A. Gigli's example

**An example of pathological gradient fields**   Gigli (2011, Example 2.4) gives a representative construction.

Consider

$$\rho(x,t) = \sum_{n=1}^{N} \delta\left(x - \exp(\mathrm{i}(2\pi n/N + \omega t))\right)$$

with $x \in \mathbb{R}^2$. This curve of densities describes $N$ points equispaced along the unit circle that rotate with angular velocity $\omega$ over time. The points are located, in radial coordinates, at $r = 1$, $\theta_n(t) = 2\pi n/N + \omega t$, where $r = |x|$ and $\theta = \mathrm{atan2}(x_2, x_1)$.

An admissible velocity field is given by $v(x) = \omega\Omega x$, with $\Omega = \begin{bmatrix} 0 & -1 \\ 1 & 0 \end{bmatrix}$: We have $\frac{\mathrm{d}}{\mathrm{d}t}x_t = v(x_t)$ for $x_t \sim \rho(t)$.

There also exists another admissible scalar field $\varphi_N$ such that $\frac{\mathrm{d}}{\mathrm{d}t}x_t = \nabla_x\varphi_N(t, x_t)$. It is given by (for example) $\varphi_N(t, x) = \omega N^{-1} r^N \sin(N(\theta - \omega t))$. Indeed, we can verify $\partial_r\varphi_N(t,x)\big|_{x=x_t} = 0$ and $\frac{1}{r}\partial_\theta\varphi_N(t,x)\big|_{x=x_t} = \omega$, i.e. $\nabla_x\varphi_N(t,x)\big|_{x=x_t} = \omega\hat{\theta} = \omega\Omega x_t$.

This field is highly oscillatory, more so as $N \to \infty$. In this limit, we should obtain a constant $\varphi_\infty$, but it is evident that $\varphi_N$ is not a sensible target of inference compared to the much more regular field $v$.

It is shown by Gigli (2011, Example 2.3) that *every* gradient vector field that is compatible with the curve $t \mapsto \rho(t)$ exhibits this behavior.

# B. Relation between curl and kinetic-energy regularizer

We know from OT theory that an admissible vector field is kinetic energy-optimal if and only if it is of gradient form, generalizing the concept of gradient in the case of regularity issues; see Ambrosio et al. (2005, Proposition 8.4.5). Furthermore, $\nabla u = (\nabla u)^T$ for every sufficiently smooth gradient field, and under the usual smoothness and simply connectedness assumptions the reverse implication is true by the Poincaré lemma (Lee, 2012, Corollary 17.15). Given the complexity of the nonlinear optimization, we do not have a numerical guarantee that the two choices lead to the same vector field, but we see in the numerical experiments that they lead to very similar sample trajectories (see Figures 2 and 4). The advantage of the kinetic-energy regularizer is that it does not require derivative information of $u$ and hence avoids backpropagation during optimization.

# C. Formal argument why gradient fields minimize the kinetic energy

We give a formal argument for why the minimal kinetic energy field takes the form of a gradient for the convenience of the reader. A similar line of reasoning can be found in (Neklyudov et al., 2023). Consider the objective

$$\min_u \max_s \left( \int \frac{1}{2}|u_t|^2 \, \mathrm{d}\rho_t(x) \, \mathrm{d}t + \int (s_t\partial_t\rho_t - \nabla s_t \cdot u_t\rho_t) \, \mathrm{d}x \, \mathrm{d}t \right).$$

The function $s$ plays the role of a Lagrange multiplier to enforce the continuity equation constraint. Assuming we can swap the order of max and min, we can explicitly solve for the minimal $u^{\min}$. Taking the first variation with respect to $u$ gives, for every admissible perturbation $\delta u$,

$$0 = \int (u_t^{\min}(x) - \nabla s_t(x)) \cdot \delta u_t(x) \, \mathrm{d}\rho_t(x) \, \mathrm{d}t \,,$$

and hence $u_t^{\min}(x) = \nabla s_t(x)$ for $\rho_t \, \mathrm{d}t$-almost every $(x,t)$. Substituting this expression back into the original objective yields

$$\max_s \int \left( \frac{1}{2}|\nabla s_t|^2\rho_t + s_t\partial_t\rho_t - \nabla s_t \cdot \nabla s_t\rho_t \right) \mathrm{d}x \, \mathrm{d}t = -\min_s \int \left( \frac{1}{2}|\nabla s_t|^2\rho_t - s_t\partial_t\rho_t \right) \mathrm{d}x \, \mathrm{d}t.$$

*Table 2.* Sensitivity of the Vlasov–Poisson two-stream electric-energy relative error to the gauge weight $\lambda$.

| Gauge | $\lambda = 1e{-}5$ | $\lambda = 1e{-}4$ | $\lambda = 1e{-}3$ | $\lambda = 1e{-}2$ |
|-------|------|------|------|------|
| DIV   | 0.065 | 0.080 | 0.065 | 0.102 |
| CURL  | 0.057 | 0.051 | 0.043 | 0.065 |
| KIN   | 0.061 | 0.074 | 0.071 | 0.062 |

# D. Details on numerical examples

## D.1. Tracer particle example

The tracer particles are advected with the flow field from the `spectral_forced_turbulence` example in the `jax-cfd` project with the default choice of parameters therein. The initial $\rho_0$ particle distribution follows a standard Gaussian centered in the $[0, 2\pi)^2$ square. Particles that exit this box re-enter according to periodic boundary conditions.

## D.2. Vlasov-Poisson problems

We follow the same problem setups given in (Berman et al., 2024). In particular, the problem is parametrized by a characteristic (Debye) length parameter $\mu$. We consider ten values of $\mu$.

The governing equation for a particle located at $x_t = [x_t^1, x_t^2]^\top$ is

$$\frac{\mathrm{d}}{\mathrm{d}t} \begin{bmatrix} x_t^1 \\ x_t^2 \end{bmatrix} = \begin{bmatrix} x_t^2 \\ \partial_{x^1} \phi_t(x_t^1) \end{bmatrix},$$

where $\phi$ solves the Poisson equation

$$-\mu^2 \Delta \phi_t = 1 - \int \rho_t(x) \, \mathrm{d}x^2 \,.$$

The parameter $\mu$ varies as $\mu_{\text{train}} \in \{1.2, 1.3, \dots, 1.9\}$ and $\mu_{\text{test}} \in \{1.25, 1.85\}$. Computed metrics are given for the average over the test set.

As a quality-of-fit metric, we use an integrated error in electric energy given by $E(t) = \frac{\mu^2}{2} \int |\partial_{x^1} \phi_t|^2 \, \mathrm{d}x^1$. We report the relative error $e_{\text{rel}} := \frac{1}{T} \int_0^T |E_{\text{true}}(t) - E_{\text{predict}}(t)| / |E_{\text{true}}(t)| \, \mathrm{d}t$. Note that the electric energy varies over two orders of magnitude throughout the simulation.

The electric potential $\phi_t$ is computed during integration of the dynamics by binning the samples in $x^1$ and solving Poisson's equation using a centered finite difference stencil.

## D.3. Sensitivity to the gauge weight $\lambda$

The penalty weight $\lambda$ in (6) sets the relative strength of the gauge regularizer compared to the weak-form distribution matching loss. In principle, this parameter controls which admissible velocity field is preferred among gauge-equivalent solutions. It should therefore be interpreted primarily as a trajectory-level modeling choice rather than as a parameter that must be finely tuned to obtain accurate marginal evolution.

To test this empirically, we swept $\lambda$ across four orders of magnitude on the Vlasov–Poisson two-stream benchmark while keeping all other training settings fixed. Table 2 reports the relative error in electric energy for the three NGIF gauges. The distributional accuracy remains stable across the sweep, indicating that careful manual tuning of $\lambda$ is not required for accurate marginal matching in this benchmark. Larger changes in $\lambda$ can still alter the learned trajectory dynamics by changing which gauge representative is preferred.

# E. Derivatives of test functions

For the standard random feature Fourier test functions with pairs

$$\phi_{2i-1}(x) = \sin(\omega_i^\top x), \qquad \phi_{2i}(x) = \cos(\omega_i^\top x),$$

their gradients are

$$\nabla\phi_{2i-1}(x) = \cos(\omega_i^\top x)\,\omega_i, \qquad \nabla\phi_{2i}(x) = -\sin(\omega_i^\top x)\,\omega_i,$$

and their Laplacians are

$$\Delta\phi_{2i-1}(x) = -\|\omega_i\|^2 \sin(\omega_i^\top x), \qquad \Delta\phi_{2i}(x) = -\|\omega_i\|^2 \cos(\omega_i^\top x).$$

# F. Algorithm

## F.1. Variant of our method building on Fokker-Planck

Instead of building on the continuity equation (4), we can also build on the Fokker-Planck equation,

$$\partial_t\rho_t + \nabla\cdot(\rho_t u_t^\varepsilon) = \frac{\varepsilon^2}{2}\Delta\rho_t.$$

In this case, the residual corresponding to the weak form for a candidate drift $v$ is

$$R_t(\phi; v) = \frac{\mathrm{d}}{\mathrm{d}t}\mathbb{E}_{\rho_t}[\phi] - \mathbb{E}_{\rho_t}[\nabla\phi\cdot v_t(x)] - \frac{\varepsilon^2}{2}\mathbb{E}_{\rho_t}[\Delta\phi],$$

under the assumption of $\nabla\rho \to 0$ as $|x| \to \infty$ for the case $x \in \mathbb{R}^d$. The parameter $\varepsilon$ is a regularization parameter. Correspondingly, instead of the flow ODE (2), one then integrates the SDE

$$\mathrm{d}x_t = u_t^\varepsilon(x_t)\,\mathrm{d}t + \varepsilon\,\mathrm{d}w_t, \qquad x_0 \sim \rho_0,$$

where $w_t$ denotes a standard Wiener process, to generate new samples. In practice we find this meaningfully helps generate accurate samples. Usually, $\varepsilon$ does not need to be tuned and can just be set to $\approx 1\mathrm{e}-2$.

---

**Algorithm 1** NGIF: Weak-form velocity inference with explicit gauge fixing

---

**INPUTS** Snapshots $\{x_k^{(i)}\}_{i=1}^N \sim \rho_{t_k}$ for $k = 0, \dots, K$; times $0 = t_0 < \cdots < t_K$; bandwidths $\{\sigma_b\}_{b=1}^B$; velocity NN $u_\theta(x, t)$; gauge $\mathcal{G}(\cdot)$; gauge weight $\lambda > 0$; loss tolerance $\epsilon_{\text{loss}} > 0$.

**SAMPLE TEST FUNCTIONS**
Choose an even number of tests $M$ and sample frequencies $\{\omega_j\}_{j=1}^{M/2}$ using the given bandwidths $\{\sigma_b\}$.
Define $\phi_{2j-1}(x) = \sin(\omega_j^\top x)$, $\phi_{2j}(x) = \cos(\omega_j^\top x)$, with $\nabla\phi_{2j-1}(x) = \cos(\omega_j^\top x)\omega_j$ and $\nabla\phi_{2j}(x) = -\sin(\omega_j^\top x)\omega_j$.

**PRECOMPUTE MOMENTS AND TIME-DERIVATIVES**
For all $k, r$: $\widehat{\mu}_{k,r} \leftarrow \frac{1}{N}\sum_{i=1}^N \phi_r(x_k^{(i)})$.
For each $r$, fit a smoothing spline $\widehat{\mu}_r(t)$ through $\{(t_k, \widehat{\mu}_{k,r})\}_{k=0}^K$ and set $\widetilde{\mu}_{k,r} \leftarrow \frac{\mathrm{d}}{\mathrm{d}t}\widehat{\mu}_r(t)\big|_{t=t_k}$ for all $k$.

**TRAINING**
**while** not converged **do**
    Sample a single time index $k \sim \text{Unif}\{0, \dots, K\}$ and a minibatch $\mathcal{B} \subset \{1, \dots, N\}$
    For all $r$: $\widehat{T}_{k,r}(\theta) \leftarrow \frac{1}{|\mathcal{B}|}\sum_{i\in\mathcal{B}} \nabla\phi_r(x_k^{(i)})^\top u_\theta(x_k^{(i)}, t_k)$
    $\ell(a, b) \leftarrow \frac{\|a-b\|^2}{\|a\|^2+\|b\|^2+\epsilon_{\text{loss}}}$
    $\mathcal{L}_k(\theta) \leftarrow \frac{1}{M}\sum_{r=1}^M \ell\left(\widetilde{\mu}_{k,r}, \widehat{T}_{k,r}(\theta)\right) + \lambda\,\mathcal{G}(u_\theta)$
    Update $\theta \leftarrow \theta - \eta\,\nabla_\theta\mathcal{L}_k(\theta)$ {e.g. Adam}
**end while**

**Output.** Learned velocity field $u_\theta(x, t)$.

---

# G. Implementation details

## G.1. Data preparation and normalization

All input variables are normalized independently to lie in $[-1, 1]$ along each dimension using statistics computed from the training set. Model training is performed in this normalized space. For evaluation, we apply the inverse normalization to

model outputs and compute all reported metrics in the original data domain. This normalization is important in practice, as it allows the same kernel bandwidths (e.g., for RFF features) to be applied consistently across dimensions and datasets.

### G.2. Computing $\dot{\mu}$ via splines

We estimate smooth moment trajectories and their time derivatives by fitting a smoothing spline to each empirical moment time series. Given observation times $t_{\mathrm{data}}$ and moment values $\mu(t_{\mathrm{data}})$, we construct $s(t) = $ `make_smoothing_spline`$(t_{\mathrm{data}}, \mu; \lambda_{\mathrm{spline}})$ (SciPy, parameter `lam`) and evaluate $\widehat{\dot{\mu}}(t_{\mathrm{data}}) = \frac{\mathrm{d}}{\mathrm{d}t}s(t)|_{t=t_{\mathrm{data}}}$. Unless otherwise stated, the spline regularization parameter is fixed to $\lambda_{\mathrm{spline}} = 10^{-5}$ in all experiments. As noted in the main text, these spline fits and derivative evaluations are precomputed once on the full dataset prior to training.

### G.3. RFF bandwidth selection

Performance is sensitive to the choice of kernel bandwidths (RFF $\sigma$ values). We therefore use a multiple-bandwidth approach, which consistently performs better than a single scale. A reasonable starting point for $\sigma$ is obtained by applying the median heuristic to the training data to obtain a base bandwidth $\sigma_{\mathrm{med}}$, and include additional scales one order of magnitude above and below, i.e., $\{\sigma_{\mathrm{med}}/10,\ \sigma_{\mathrm{med}},\ 10\,\sigma_{\mathrm{med}}\}$.

### G.4. Hyperparameters

For all experiments unless otherwise noted the following algorithmic hyperparameters were used:

|  | Gigli | Tracer particles | Vlasov–Poisson | High-dimensional turbulence |
|---|---|---|---|---|
| $\sigma_{\min}$ | 0.05 | 0.05 | 0.05 | 4.0 |
| $\sigma_{\max}$ | 0.05 | 1.0 | 1.0 | 16.0 |
| $B$ | 1 | 100 | 100 | 100 |
| $\lambda_{\mathrm{spline}}$ | $10^{-5}$ | $10^{-5}$ | $10^{-5}$ | $10^{-5}$ |
| $\varepsilon$ | 0.0 | $10^{-2}$ | $10^{-2}$ | 0.0 |
| $M$ | 50,000 | 50,000 | 50,000 | 50,000 |

|  | Tracer particles | VP (two-stream, bump) | VP (strong-Landau) | High-dimensional turbulence |
|---|---|---|---|---|
| $\lambda_{kin}$ | 1e−2 | 1e−2 | 1e−2 | 1e−2 |
| $\lambda_{curl}$ | 1e−2 | 1e−3 | 5e−3 | – |
| $\lambda_{div}$ | 1e−2 | 1e−3 | 5e−3 | – |

| Hyperparameter | Value |
|---|---|
| Learning rate (lr) | $5 \times 10^{-4}$ |
| Iterations (iters) | 500,000 |
| Scheduler | cos |
| Optimizer | Adam |
| Precision | float32 |

### G.5. Neural Net Architectures

**Autoencoder** *Architecture.* Convolutional encoder/decoder with Swish + GroupNorm (8 groups) and dropout 0.05. Four downsampling stages $128 \rightarrow 64 \rightarrow 32 \rightarrow 16$ (strided conv) mirrored by four upsampling stages (transpose conv). Base channels 32, doubling each downsample (bottleneck 512). The latent space has size $16\times16\times1$, with posterior $(\mu, \log \sigma^2) \in \mathbb{R}^{16\times16\times1}$; $\log \sigma^2$ clipped to $[-12, 6]$.

*Loss (KL-regularized).* Reconstruction $=$ MSE $+ 0.10\,$MAE plus $\beta$-KL regularization with $\beta = 5 \times 10^{-4}$. Linear KL warmup over 5000 optimizer steps.

*Latent post-processing.* After training, per-dimension whitening statistics of $\mu$ are computed over the dataset; encoding uses $\mu$ by default and returns whitened latents optionally $\tanh$-squashed to $[-1, 1]$ (enabled; scale 2.0). Decoding inverts squash (atanh) and whitening.

For all other decisions we use standard, best-practice choices for convolutional VAE training and evaluation.

**MLP Backbone**    All parameterizations of $u_\theta$ except the high-dimensional ones are done using an MLP.

For periodic problems, we embed $x$ using harmonic features with $H = 4$:

$$h(x) = [\sin(\omega_0 m x), \cos(\omega_0 m x)]_{m=1}^4, \quad \omega_0 = \tfrac{2\pi}{\text{period}},$$

and concatenate features across dimensions. For non-periodic problems, the raw input is used.

The network optionally conditions on time $t$ and auxiliary variables $\mu$. Each is embedded by a two-layer MLP with hidden width equal to the main network width and GELU activations, producing vectors in $\mathbb{R}^W$. If both are present, the embeddings are concatenated. If only one is present, it is used directly.

The backbone consists of $L$ fully connected layers of width $W$ with GELU activations. At each layer, a conditioning bias is added. The bias is produced by passing the conditioning vector through a separate two-layer MLP of width $W$ and adding the result to the pre-activation output.

Unless otherwise stated, we use: $W = 196$, $L = 7$, GELU activations, bias terms in all layers, period set per task, and task-dependent output dimension.

**UNet backbone**    For the high-dimensional experiments we employ a UNet. We follow the encoder-decoder UNet with skip connections. All architecture decisions are standard. We use GroupNorm, no dropout, and stride-2 $3\times3$ convolution. At each spatial scale for our residual blocks the channel depths are $[128, 128, 360]$.

## H. Metrics

### H.1. Total variation distance between binned particle measures

Let $\{\mathcal{B}_k\}_{k=1}^{N_{\text{bins}}}$ be a partition of $\mathbb{T}^2 = [0, 2\pi)^2$ into $N_{\text{bins}} = B_x B_y$ equal bins. Given particles $\{x_n\}_{n=1}^N$, define counts and normalized histogram

$$c_k = \sum_{n=1}^N \mathbf{1}\{x_n \in \mathcal{B}_k\}, \qquad p_k = \frac{c_k}{\sum_{\ell=1}^{N_{\text{bins}}} c_\ell}.$$

For two particle sets with histograms $p, q \in \mathbb{R}^{N_{\text{bins}}}$, the total variation distance is

$$d_{\text{TV}}(p, q) = \frac{1}{2} \sum_{k=1}^{N_{\text{bins}}} |p_k - q_k| = \frac{1}{2}\|p - q\|_1.$$

