# OpenReview forum: "Leveraging Gauge Freedom for Learning Non-Gradient Population Dynamics of Stochastic Systems"
_ICML.cc/2026/Conference — ICML 2026 regular_

### Official Review · Reviewer_sJpn · 2026-03-07

**Soundness:** 3
**Presentation:** 3
**Significance:** 2
**Originality:** 2
**Overall Recommendation:** 4
**Confidence:** 4

**Summary:**

The paper considers the problem of learning population dynamics from data that does not contain information about single trajectories (which could be meaningless in stochastic dynamics) but rather independent samples at different time points. Most approaches in the literature resolve the ambiguity of the velocity field in the physical space by requiring the velocity to provide minimal kinetic energy, which in practice is equivalent to assume the velocity to be the gradient of some scalar potential. Instead, the authors propose to enforce the continuity equation in a weak form in the sense of distributions, i.e. over test functions. While this requires choosing ad-hoc which and how many test functions to use. It provides freedom in selecting a bias for the velocity field by means of a separate penalty term. Depending on the penalty term, the velocity field can be biased towards, for instance, minimal kinetic energy or divergence free.

**Compliance With Llm Reviewing Policy:**

Affirmed.

**Final Justification:**

I believe that the proposed approach provides an elegant solution to problems with non-gradient fields, leaving freedom to choose a prior in the form of a penalty term. I kept my score to Weak accept because within the rebuttal my concerns about scalability for dimensionality (and partially for the number of test functions) were not fully addressed.

**Key Questions For Authors:**

### Questions
1. Is the code for this paper available online (anonymously)?
2. Figure 2: did you compare against other methods, e.g. those used in Experiment 3.3 such as HOAM, DICE, AM? I understand that the point of the figure is to show the effect of the different gauges. However, if the point is not just illustrative, a comparison would be needed, and would also help understand if the proposed method would be relevant for divergence free settings as well
3. The intended scope of the proposed approach is not entirely clear to me. From a theoretical perspective I understand that not relying on a gradient flow makes it suitable to settings where the velocity has a strong rotational component. I also understand that other penalty are suggested and possible. However, in which other settings, apart from rotational velocity field, would you expect the proposed approach to have a significant advantage over competing methods?
4. Could you provide a quantitative study on the choice of test functions and how their number affects the accuracy/performance? I would be particularly interested in the scaling with the dimensionality as well
5. Experiment 3.4 relies on a latent space, which doesn't really address the high-dimensionality scaling. What is the largest dimensionality you were able to treat without resorting on a latent space?
6. Why are the trajectories in the three right-most plots significantly shorter?

### Typos (“l” means left column and “r” means right column)
- line 106 (l): citation should be outside of parenthesis
- line 100 (r): citation should be outside of parenthesis
- line 170 (l): full stop after the equation missing
- line 270 (l): citation should be outside of parenthesis
- line 143 (r): remove full stop after the equation
- line 280 (l): pre-compute → pre-computed
- line 304 (l): remove the
- line 307 (r): add space after Figure 2
- line 318 (r): obtain → obtained
- line 328 (r): citation should be outside of parenthesis
- line 343 (l): citation should be outside of parenthesis
- line 360 (r): see 1 → see Section 1
- $\frac{\text{d}}{\text{d}t}$ and $\frac{d}{dt}$ are used interchangeably, see equation 3, 4, line 146 right, equation line 157)

**Limitations:**

yes

**Strengths And Weaknesses:**

### Strengths:
1. The paper is very well written and easy to follow
2. The proposed method is novel and provides more freedom than standard approaches relying on gradient flows, The freedom comes in terms of choice of underlying inductive bias, which is enforced as an additional regularization/penalty term

### Weaknesses:
1. The proposed approach has a comparative advantage specifically in settings where the underlying vector field has a significant rotational component. Even though some other penalties for gauge freedom are possible, the behaviour in other circumstances is not explored as systematically.
2. Reproducibility. At the present state the paper does not provide code of their implementation. Some details concerning experiments and datasets are contained in the appendix, but without an exhaustive description that could replace providing the code.
3. Scalability to high dimensionality: the method relies on weakly enforcing the continuity equation in the distribution sense by using test functions. Therefore, results depend on which and how many test functions are used, which probably doesn't scale well with the dimensionality. One high dimensional problem is provided, but it is actually learned in the latent space.

Overall, I believe the paper to be interesting for the ICML community and I would lean towards full acceptance if (i) the intended use cases of the proposed approaches were made clearer (i.e. where the authors expect the proposed approach to be useful besides rotational velocity fields) and if concerns about (ii) scaling to (non-latent) high-dimensional settings and about (iii) the behaviour of test functions was resolved through a quantitative analysis. I comment more thoroughly on such points in the questions section.

---

> ### Author Rebuttal · Authors · 2026-03-30
>
> We thank Reviewer sJpn for their thorough review. We especially appreciate the clear articulation of what would move them toward full acceptance. We address each point below.
>
> —
>
> ***“[...] (i) intended use cases of the proposed approaches were made clearer [...] ”***
>
> We call out here two more advantages:
>
> NGIF avoids the representational and optimization difficulties associated with gradient-based parameterizations. Hard-enforcing a gradient constraint can make training substantially more difficult. As the Gigli example shows, even simple non-gradient behavior may need to be represented through pathological or highly oscillatory potentials. This is shown further in our superior performance to competing methods such as HOAM, DICE, and AM and confirmed by our new experiments evaluating these methods on the tracer particles and high-dim turbulence.
>
> Additionally, NGIF is well suited to settings where one has prior physical knowledge about trajectories. This is often the case in physics problems where one has a rough idea how the particles should behave (e.g., particles corresponding to incompressible fluid). In such cases, NGIF does not merely offer greater expressivity than gradient-restricted methods; it also allows this prior to be imposed at all.
>
> In our revision, we will add a clear three-item list to the Introduction to emphasize the benefits of this method: (a) capture non-gradient (e.g., rotational) dynamics, (b) ability to impose physics prior on trajectory dynamics, and (c) avoiding training pathologies and costly backprop through spatial derivatives.
>
> ***"(ii) scaling to (non-latent) high-dimensional settings [...] (iii) [...] behaviour of test functions was resolved through a quantitative analysis" Experiment 3.4 relies on a latent space, [...] largest dimensionality you were able to treat without resorting on a latent space?"***
>
> We agree with the reviewer that scaling with dimension is an important practical consideration. We conducted two additional experiments.
>
> First, we consider a synthetic time-dependent Gaussian benchmark up to dimension $d = 64$. At each time, the target distribution is a single Gaussian with fixed isotropic covariance and a mean that evolves smoothly, with each two-dimensional coordinate pair following a circular trajectory with a pair-dependent phase shift. As the dimension increases, the problem becomes harder because the dynamics must be resolved simultaneously across a growing number of coordinate pairs. The reported relative error in the mean shows that in low dim, performance is already stable with a modest number of test functions $M = 5 \times 10^3$, whereas in higher dimensions increasing the number of test functions to $M = 5 \times 10^4$ improves accuracy substantially.
>
> | dim | M=500 | M=5000 | M=50000 |
> |-|-|-|-|
> | 2 | 0.12 | 0.12 | 0.12 |
> | 32 | 0.73 | 0.37 | 0.16 |
> | 64 | 0.57 | 0.31 | 0.24 |
>
> Second, we consider another new example, the 2D linear advection equation $\partial_t u + c_x(t) \partial_x u + c_y(t) \partial_y u = 0$ on $[0,1]^2$ with periodic boundary conditions. The y-velocity is fixed at $c_y = 1$, while $c_x(t)$ is a Gaussian random variable. The domain is discretized on a $64 \times 64$ spatial grid. Each sample thus is a $d = 4,096$ dimensional state vector representing the initial condition (Gaussian bump) transported via linear advection. Using only $50,000$ test function our NGIF accurately captures this stochastic lateral drift, achieving a relative error of 0.12 in the estimated y-velocity. By contrast DICE, HOAM, and AM are unable to learn on such high dimensional problems.
>
> As noted in our Limitations section, random Fourier test functions are inexpensive and perform well in our experiments, but their informativeness can degrade in very high dimensions. That said, both the original and new results confirm the method remains effective well beyond low-dimensional settings.
>
> ***"Figure 2: did you compare against other methods [...]?"***
>
> Based on the reviewer’s comment, we ran HOAM, DICE, AM on this experiment. We report the total variation distance (see Figure 3) in the table:
>
> | | NGIF-div  | NGIF-curl | NGIF-kin  | grad-flow | DICE | HOAM | AM |
> |-|-|-|-|-|-|-|-|
> | final T.V.D | 0.51 | 0.48 | 0.49 | 0.62 | 0.61|  0.57 | 0.72 |
>
> These new results agree with our conclusion that gradient parameterization can be overly restrictive.
>
> ***"Why are the trajectories [...] significantly shorter?"***
> - Trajectory lengths reflect the learned velocity magnitudes. The kinetic/curl regularizers and gradient flow penalize speed or rotation, producing slower transport and shorter paths. The divergence regularizer does not, so its trajectories match the true dynamics more closely in both length and character.
>
> ***"Is the code available?"***
> - We will release all code and data upon acceptance, including training scripts, data generation, and evaluation pipelines.
>
> ***Typos***
> - Thank you for the detailed list - all typos will be corrected.

---

> > ### Author Rebuttal · Reviewer_sJpn · 2026-04-03
> >
> > I would like to thank the authors for their detailed rebuttal.
> >
> > Concerning (i), I believe that with the listed application setting the motivation for the proposed approach can be better understood. Current experimental evaluation, however, supports mostly the first motivation about non-gradient fields. Concerning (ii)/(iii), I thank the authors for the additional experiments. While the time-dependent Gaussian experiments does provide some intuition about scaling, from only three points it's hard to extrapolate the behavior. From what I can see 2D problems can be solved with few test functions while already for 32 and 64 two orders of magnitude more are needed. I would be curious how this translates to e.g. $d=100$ and $d=1000$. With the new advection equation the dimensionality is indeed increased this further so it would be interesting to see a consistent behavior for the time-dependent Gaussian experiment as well. Lastly, I would like to note that availability of the code at the time of the rebuttal, despite not being mandatory, would have been helpful to better assess reproducibility.
> >
> > I would like to thank the authors again. Given the concerns stated above I will keep my recommendation to a weak accept.

---

> > > ### Author Response · Authors · 2026-04-06
> > >
> > > Thank you again for the thoughtful follow-up and for the positive overall assessment. We appreciate the reviewer’s continued engagement.
> > >
> > > On (i), we agree that this point is important. And we reiterate that our revision will focus on making the following points clearer: (a) we can capture rotational dynamics, (b) we give the ability to impose physics prior on trajectory dynamics, and (c) we avoid training pathologies imposed by hard gradient parameterizations.
> > >
> > > We will also emphasize in the revision that two experiments demonstrate the ability to impose physics priors on trajectory dynamics. First, we show this in the Vlasov example, where the choice of gauge regularizer motivated by incompressible flow dynamics encodes meaningful prior knowledge about the trajectory dynamics while maintaining strong distributional accuracy. Likewise, in the tracer setting, the divergence regularizer corresponds to an incompressibility prior, and we demonstrate through the trajectory plots that this yields trajectories that are qualitatively closest to the physical ones.
> > >
> > > On (ii)/(iii), we agree that the time-dependent Gaussian experiment should not be over-interpreted as a full scaling law. The new 2D linear advection PDE experiment may be more informative. It shows our behavior on a more realistic high-dimensional problem. There, we treat the problem in native dimension d = 4096 on a 64 x 64 grid, and NGIF remains accurate.
> > >
> > > We also note as stated in our response to Reviewer tFXm, increasing the number of test functions from 25,000 to 200,000 increases training runtime by only about 20%, which suggests that in this regime the dominant cost is the backbone model rather than the test-function evaluations themselves.
> > >
> > > Lastly, we think a major take away from the paper is that for truly high-dimensional data, NGIF composes naturally with latent embeddings, as demonstrated in our turbulence experiment (Section 3.4) where we effectively treat 128x128 dimensional data via an autoencoder. Learning models in a latent space has become a standard practice in machine learning and for us, it unlocks genuinely high-dimensional problems for our approach.
> > >
> > > We state clearly in the paper and the rebuttal that random Fourier test functions can become less informative in very high dimensions. However, at the same time, we believe the current evidence demonstrates that NGIF already scales to flow problems in practically relevant high-dimensional settings.
> > >
> > > We thank the reviewer for pressing these points, and we hope that the clarifications help address the remaining concerns.

---

### Official Review · Reviewer_tFXm · 2026-03-11

**Soundness:** 3
**Presentation:** 3
**Significance:** 2
**Originality:** 3
**Overall Recommendation:** 4
**Confidence:** 4

**Summary:**

This paper points out that, based on the form of the continuity equation, there are actually multiple different velocity fields that can cause a distribution to evolve in the same way. However, most existing methods for learning distributions from data, such as those based on optimal transport or Schrödinger bridges, typically constrain the velocity field to be a gradient field. These methods often perform poorly when dealing with systems like fluids with vorticity. This paper utilizes various moments of the probability distribution to constrain the distribution matching, introducing explicit regularization terms to guide the model in learning non-gradient velocity fields that conform to specific physical priors. Experiments demonstrate that this method outperforms various baselines in learning dynamics across diverse non-gradient systems.

**Compliance With Llm Reviewing Policy:**

Affirmed.

**Final Justification:**

According to the continuity equation, many different velocity fields yield the same evolution of $p_t(x)$. The authors wish to avoid the gauge $v(x,t) = \nabla s(x,t)$ found in problems like optimal transport (because it cannot handle vortices in fluids). In the rebuttal, the authors demonstrated that their algorithm is robust to the parameter $\lambda$, and that a more accurate evolution of the distribution can be obtained by increasing the number of test functions. Based on this, I am raising my score to 4.

**Key Questions For Authors:**

1. Could the authors please clarify the strategy for selecting $\lambda$ in the loss function? Additionally, how are the regularization terms chosen? Is it possible to select the regularization terms based on the underlying characteristics of the data?
2. For extremely high-dimensional data, is it necessary to adjust the number of test functions? If so, how should this adjustment be made, and how would it impact the overall computational cost? Have the authors evaluated this method on high-dimensional data (e.g., 50 dimensions)? If so, what were the experimental results?

**Limitations:**

yes

**Strengths And Weaknesses:**

### Strengths
1. By allowing the existence of non-gradient fields (thereby removing the inductive bias present in previous methods), the model can accurately capture the rotational and incompressible characteristics commonly found in fluid dynamics and Hamiltonian systems.
2. The proposed method is computationally efficient. The computational complexity of the distribution matching loss grows linearly with the dimensionality \(d\), and it achieves results surpassing the baselines on multiple datasets.

### Weaknesses
1. The weights of the distribution matching loss and the regularization loss within the total loss require manual tuning.
2. As the dimensionality increases, a growing number of test functions may be required to constrain the distribution matching, which will inevitably lead to higher computational costs

---

> ### Author Rebuttal · Authors · 2026-03-30
>
> We thank Reviewer tFXm for their concise and focused review. The λ sensitivity and dimensional scaling, are important questions, and we are glad to be able to address them with concrete new experiments below.
>
> —
>
> ***"The weights [...] require manual tuning. Could the authors please clarify the strategy for selecting λ in the loss function?"***
>
> Our NGIF approach is fairly robust against this trade-off given by λ when it comes to matching the distributional evolution. We want to emphasize that careful manual tuning of λ is not required for accurate marginal matching. The key insight is that λ mostly controls which admissible velocity field among gauge-equivalent solutions is returned. To demonstrate this concretely, we swept λ across 4 orders of magnitude (1e-5 to 1e-2) on the two-stream Vlasov-Poisson benchmark. The distributional accuracy (relative error in electric energy) remains stable throughout:
>
> | | λ=1e-5  | λ=1e-4  | λ=1e-3  | λ=1e-2  |
> |-|-|-|-|-|
> | div  | 0.065 | 0.080 | 0.065 | 0.102 |
> | curl | 0.057 | 0.051 | 0.043 | 0.065 |
> | kin  | 0.061 | 0.074 | 0.071 | 0.062 |
>
> This indicates that the population-level prediction is robust to the precise choice of $\lambda$ over a broad range, even though λ does control which admissible trajectory dynamics are preferred.
>
> ***"Is it possible to select the regularization terms based on [...] data?"***
>
> Yes, the regularizer can be selected based on prior knowledge of the expected physics. Each encodes a concrete structural prior. For example $G_{\mathrm{DIV}}$ promotes approximately incompressible transport. Thus on the tracer-particle benchmark where the ground-truth field is divergence-free, $G_{\mathrm{DIV}}$ yields trajectories qualitatively closest to the physical ones.
>
> ***"For extremely high-dimensional data, is it necessary to adjust the number of test functions? [...] Have the authors evaluated this method on high-dimensional data (e.g., 50 dimensions)?"***
>
> First we note that the latent space of the auto-encoder is 64 dim. Thus we have already had success on problems above 50 dim. We will make this clear in our revision.
>
> Beyond this we conduct two additional experiments.
>
> First, we consider a synthetic time-dependent Gaussian benchmark up to dimension $d = 64$. At each time, the target distribution is a single Gaussian with fixed isotropic covariance and a mean that evolves smoothly, with each two-dimensional coordinate pair following a circular trajectory with a pair-dependent phase shift. As the dimension increases, the problem becomes harder because the dynamics must be resolved simultaneously across a growing number of coordinate pairs. The reported relative error in the mean shows that in low dim, performance is already stable with a modest number of test functions $M = 5 \times 10^3$, whereas in higher dimensions increasing the number of test functions to $M = 5 \times 10^4$ improves accuracy substantially.
>
> | dim | M=500 | M=5000 | M=50000 |
> |-|-|-|-|
> | 2 | 0.12 | 0.12 | 0.12 |
> | 32 | 0.73 | 0.37 | 0.16 |
> | 64 | 0.57 | 0.31 | 0.24 |
>
> Second, we consider another new example, the 2D linear advection equation $\partial_t u + c_x(t) \partial_x u + c_y(t) \partial_y u = 0$ on $[0,1]^2$ with periodic boundary conditions. The y-velocity is fixed at $c_y = 1$, while $c_x(t)$ is a Gaussian random variable. The domain is discretized on a $64 \times 64$ spatial grid. Each sample thus is a $d = 4,096$ dimensional state vector representing the initial condition (Gaussian bump) transported via linear advection. Using only $50,000$ test function our NGIF accurately captures this stochastic lateral drift, achieving a relative error of 0.12 in the estimated y-velocity. By contrast DICE, HOAM, and AM are unable to learn on such high dimensional problems.
>
> With regards to the comp. costs of additional test functions we also report the total wall clock train time for the 2D linear advection as we increase the number of test functions.
>
> | | M=25,000 | M=50,000 | M=100,000 | M=200,000 |
> |-|-|-|-|-|
> | train time (min.) | 282    | 289    | 306     | 336     |
>
> From 25,000 to 200,000, we only increase the wall clock time by ~20%. The take away is that the computation of the test functions is computationally cheap relative to the cost of evaluating and backproping through the UNet for a high dimensional problem.
>
> As noted in our Limitations section, random Fourier test functions are inexpensive and perform well in our experiments, but their informativeness can degrade in very high dimensions. That said, **both the original and new results confirm the method remains effective well beyond low-dimensional settings**. For truly high-dimensional data, NGIF composes naturally with latent embeddings, as demonstrated in our turbulence experiment (Section 3.4, ambient dimension $128 \times 128$). We will make this point more explicit in the revision.

---

> > ### Author Rebuttal · Reviewer_tFXm · 2026-04-02
> >
> > I thank the authors for their detailed response. I believe my concerns have been addressed, and I will raise my score to 4.

---

> > > ### Author Response · Authors · 2026-04-02
> > >
> > > We sincerely thank the reviewer for their engagement throughout the discussion and for raising their score. The exchange helped us sharpen the presentation of key aspects of our method, and we believe the paper is stronger as a result.

---

### Official Review · Reviewer_DoVi · 2026-03-12

**Soundness:** 3
**Presentation:** 3
**Significance:** 4
**Originality:** 3
**Overall Recommendation:** 5
**Confidence:** 4

**Summary:**

This paper proposes a method for population dynamics inference, termed Non-Gradient Inference Flows (NGIF), for learning non-gradient velocity fields from time-marginal snapshot data. While many existing approaches restrict the inferred field to a gradient form, corresponding to a minimal-kinetic-energy gauge, the authors argue that this is only one possible gauge choice. Their method instead separates marginal matching from velocity-field selection, and uses explicit gauge regularizers in Eqs. (7)–(9) to encode alternative structural biases. Experiments on several physical systems suggest that this broader formulation can outperform gradient-restricted baselines in settings with rotational structure.

**Compliance With Llm Reviewing Policy:**

Affirmed.

**Final Justification:**

While the proposed regularizers effectively narrow the solution space, a fundamental limitation remains: there can still be numerous gauge-equivalent fields that achieve the same minimum loss. Consequently, the final selection among these fields inevitably relies on the implicit bias of the neural network. I acknowledge that this is an inherent difficulty of the problem itself, which the authors have also discussed in the paper.

Although the paper lacks a definitive recipe for resolving these exact ties without relying on network bias, the novelty and practical utility of the framework are evident. Given the authors' sincere and thorough rebuttal, I am raising my score to Accept.

**Key Questions For Authors:**

**Major**
- (Weak continuity equation) The weak continuity formulation is well motivated in the sample-based setting, since directly estimating $\rho_t$ and especially its spatial/temporal derivatives from snapshots can introduce substantial numerical error. However, I wonder whether the benefit of the proposed approach may extend beyond this setting. Even when the population $\rho_t$ itself is directly measurable, enforcing the strong-form continuity equation would still require differentiating $\rho_t$, which may remain numerically unfavorable. In that regime, does the weak formulation still offer practical advantages? Have the authors tested the method in a setting where $\rho_t$ is explicitly available, to assess whether avoiding density derivatives is beneficial even then?
- (Hyperparameter tuning) The core weakness of this method is that the inferred vector field largely affects the trade-off between the weak continuity equation loss and the gauge regularizer and the non-uniqueness of the solution. The hyperparameter $\lambda$ is therefore very important, but I am not sure how to tune this hyperparameter. How were $\lambda$ and the test function hyperparameters selected in practice? Is there a validation strategy that does not rely on access to the ground-truth dynamics?
- (High-dimensional simulation) Figure 5 does not include baseline comparisons, which makes it hard to judge whether the proposed method actually improves over prior approaches in the high-dimensional setting. Could the authors add at least one baseline, or clarify why such a comparison is infeasible?
- (Reproducibility) I encourage the authors to release code and data to improve reproducibility.

**Minor**
- Please restate more explicitly in the formal setup paragraph that the observations consist of independent samples from each time marginal and that trajectory correspondence is not observed.

**Limitations:**

Please refer to the Weaknesses section above for a more detailed discussion. In summary, the primary limitation of this work is that the method cannot guarantee the uniqueness of the solution, and it is therefore difficult to determine which solution is the most appropriate. In addition, the hyperparameter $\lambda$ can significantly affect the results.

**Strengths And Weaknesses:**

**Strengths**
- The paper addresses an interesting and timely problem: inferring population-level dynamics when non-gradient structure may be important. This is especially timely in light of recent interest in odd dynamics, such as odd viscosity and chirality, where rotational and non-gradient effects play a central role.
- The paper is clearly written, and the motivation is easy to follow. Figure 1 is particularly effective at illustrating why enforcing a gradient-field parameterization can lead to an unnecessarily irregular target, even when the underlying dynamics are simple.


**Weakness**
- While I agree with the practical applicability of this method, I am not sure of the theoretical robustness. The method remains underdetermined by construction, and the final inferred field depends substantially on the chosen gauge regularizer and its hyperparameter $\lambda$. While this flexibility is part of the appeal of the framework, it also makes the result strongly dependent on modeling choices.
- More generally, the practical objective is based on finitely many test functions and a soft trade-off between weak continuity matching and gauge regularization, rather than an exact constrained formulation. This leaves some uncertainty about identifiability and robustness.

---

> ### Author Rebuttal · Authors · 2026-03-30
>
> We thank Reviewer DoVi for their thoughtful review and for highlighting the significance and timeliness of the problem.
>
> ---
>
> ***"While I agree with the practical applicability of this method [...] method remains underdetermined by construction, and the final inferred field depends substantially on the chosen gauge regularizer and its hyperparameter. [...]"***
>
> We appreciate this point and believe it helps to separate two notions of "underdeterminedness." The first is intrinsic: from time-marginal data alone, the velocity field is determined only up to gauge-equivalent components, since the continuity equation depends on $u_t$ only through $\nabla \cdot (\rho_t u_t)$. This is a property of the problem, not our method. What is identifiable from the weak continuity formulation is the population dynamics at the level of marginal evolution; the regularizer selects representative trajectory dynamics within the admissible equivalence class. We view the opening of this design space as a key contribution of our work.
>
> The second is practical: finitely many test functions and a soft penalty replace the exact formulation, introducing the usual trade-off familiar from regularized ML. In principle, $\lambda$ can affect both how closely the marginal residual is minimized and which velocity field is selected. However, we see that our NGIF approach is fairly robust against this trade-off when it comes to matching the distributional evolution. To provide further evidence for this, we ran a new experiment on the two stream Vlasov problem where we vary the strength of lambda across 4 orders of magnitude, $10^{-5}, 10^{-4}, 10^{-3}, 10^{-2}$. We find that the performance in terms of relative error in electric energy (a distributional quantity, i.e., it depends on the evolution of the distribution only but not on the trajectory dynamics) is quite stable:
>
> | | 1e-5  | 1e-4  | 1e-3  | 1e-2  |
> |-|-|-|-|-|
> | div  | 0.065 | 0.080 | 0.065 | 0.102 |
> | curl | 0.057 | 0.051 | 0.043 | 0.065 |
> | kin  | 0.061 | 0.074 | 0.071 | 0.062 |
>
> This confirms that distributional accuracy is robust across a wide range of $\lambda$; what changes is the character of the learned trajectories, which is precisely the gauge freedom our paper exposes.
>
>
> ***"How were $\lambda$ and the test function hyperparameters selected in practice?"***
>
> In practice, they are selected based on a validation set using only population-level information available from the data (e.g. held-out time-marginal samples and derived QoIs). Thus, validation is performed at the level of marginal evolution, which is the object our method is designed to match, rather than at the level of unobserved trajectories, for which NGIF offers the additional, regularizer-guided modeling choice. We will make this transparent in the revision.
>
>
> ***"Figure 5 does not include baseline comparisons, which makes it hard to judge whether the proposed method actually improves over prior approaches in the high-dimensional setting."***
>
> This is an important point. In response we run HOAM, DICE, and AM on our high dimensional turbulence and compute the relative error in the enstrophy over time. The results are as follows:
>
> | | NGIF | DICE | HOAM | AM   |
> |-|-|-|-|-|
> | avg enstrophy err | 0.09 | 0.71 | 0.67 | 0.69 |
>
> Both of these new results agree with our NGIF-based grad-flow method, providing further evidence that constraining dynamics to a gradient parameterization can be overly restrictive and is hard to train (as discussed in Appendix C of Neklyudov et al).
>
> ***"Even when the population $\rho$ itself is directly measurable, enforcing the strong-form continuity equation would still require differentiating $\rho$... does the weak formulation still offer practical advantages? [...]"***
>
> This is an insightful comment. Yes, we think the weak form formulation still offers practical advantages even when $\rho_t$ is explicitly available. Exactly as the reviewer points out, the weak formulation shifts the time derivative onto the test functions, for which the time derivative can often be computed more accurately (and even in analytical form) than for the density function. For example, when $\rho_t$ is available as a function to evaluate, back-propagation to obtain derivative information can still be restrictively expensive.
>
> In this work, we only focus on the setting that we only have samples available. It remains an interesting direction for future work to consider settings where $\rho_t$ is given and one still would like to learn a velocity field.
>
> ***"I encourage the authors to release code and data to improve reproducibility."***
>
> We will release all code and data upon acceptance. A complete reproduction package including training scripts, data generation, and evaluation will be made publicly available.
>
> ***"Please restate more explicitly in the formal setup […] correspondence is not observed."***
>
> We will add this clarification to the setup paragraph in the revision. Thank you for the suggestion.

---

> > ### Author Rebuttal · Reviewer_DoVi · 2026-04-02
> >
> > I appreciate the authors’ efforts in addressing my concerns. In particular, the additional baseline experiments in the high-dimensional setting are valuable and strengthen the empirical support for the paper by demonstrating the advantage of the proposed method in that regime.
> >
> > However, my main concern regarding how to choose an appropriate $\lambda$ and, more broadly, how to select a plausible resulting vector field is not fully resolved. The newly added robustness analysis over varying $\lambda$ is helpful, and it suggests that distributional accuracy remains stable across a reasonably wide range. Nevertheless, I am still uncertain how, in practice, one should use a validation set to choose among different vector fields that may all achieve similar distributional accuracy. I understand that this is inherently difficult in the absence of trajectory-level supervision, but if the method is to be broadly applicable, clearer guidance on model selection and on choosing among admissible vector fields would be important.
> >
> > Overall, I appreciate the additional evidence and clarifications, and I will maintain my positive score.

---

> > > ### Author Response · Authors · 2026-04-02
> > >
> > > We thank the reviewer for this follow-up and apologize for the ambiguity in our initial response. We think we now understand the concern more precisely, and it helps to separate two distinct ways of conducting validation.
> > >
> > > **Distributional validation (selecting λ).** Given a choice of regularizer, we select λ by holding out time-marginal samples and evaluating a distributional metric on these held-out snapshots. Note that if we only have time-marginal samples (i.e., no trajectories), then a distributional metric is the only quantity we can validate on. This is what we do in practice across all experiments, and it is analogous to selecting a regularization weight in any penalized regression problem. Our λ sweep confirms this selection is forgiving. We will describe this procedure more thoroughly in a new appendix section.
> > >
> > > **Trajectory validation (choosing the regularizer).** Selecting among gauge-equivalent velocity fields that all achieve similar distributional accuracy cannot be done from marginal data alone; it requires some form of trajectory-level knowledge. When such knowledge is available, our framework can incorporate it directly as a gauge regularizer. For example, suppose one knows the true velocity field is not exactly divergence-free but has a known spatiotemporal divergence profile $d(x,t)$. One can define a modified regularizer $\mathcal{G}(v) = \|\nabla \cdot v(x,t) - d(x,t)\|^2$ and use it as the gauge penalty. This works precisely because such structural information lives in the gauge-free component. Moreover, adherence to this gauge penalty could provide an additional criterion for selecting λ via validation as well. Indeed, our $\mathcal{G}_{\mathrm{DIV}}$ tracer experiment in Section 3.2 is simply the case where $d(x,t)$ is uniformly zero. This illustrates that when trajectory-level observables are available, they can be incorporated directly as gauge regularizers in our framework.
> > >
> > > We thank the reviewer again for this productive exchange; we feel this discussion has been very helpful and we will incorporate all their comments in our revision.

---

### Official Review · Reviewer_vAsx · 2026-03-13

**Soundness:** 3
**Presentation:** 3
**Significance:** 3
**Originality:** 2
**Overall Recommendation:** 4
**Confidence:** 4

**Summary:**

The paper focuses on **inference of population dynamics from time marginals** for settings with non-gradient flow fields. The authors mention that observed time marginal densities do not determine uniquely the underlying velocity field, since from the continuity equation the constraint is only on the weighted divergence of the probability flux $\nabla \cdot (\rho_t u_t)$ and not the velocity field $u_t$ itself.
The main point the authors bring forward is that existing approaches for identifying population dynamics largely fix the undetermination of the continuity equation by hard-coding the minimal-kinetic-energy gauge, which forces gradient fields. Instead one can take separate steps for (i) satisfying the weak continuity equation from (ii) choosing a field within the equivalence class by an explicit regularize.

The authors propose a method that enforces the weak continuity with random Fourier feature test functions, and consequently add gauge regularization (3 variants: with kinetic energy penalty, curl penalty, divergence penalty). The authors claim that this approach improves distributional accuracy when the underlying dynamics contain a significant non-gradient structure.

The paper is interesting and up to some mathematical fixes and polishing I think it will be a good addition to the conference.

**Compliance With Llm Reviewing Policy:**

Affirmed.

**Final Justification:**

I would like to thank again the authors for their responses. Their rebuttal reinforced my assessment.

My main concern, that the authors acknowledged in their responses, is that there is no practical way proposed yet on how to select which of the well-performing admissible velocity fields is actually closest to the ground-truth field. Thus I propose that they should include this in their discussion together with some insights/speculations on how to use heuristics to determine the exact ground truth.

**Key Questions For Authors:**

- Can the authors better justify the trade-off formulation versus the constrained formulation.
- How sensitive is the method to the exact choice of the test function family?

- The authors only briefly mention in the introduction that an alternative approach to fix the undetermination of the underlying flow field is to account for the non-trivial geometric/metric structure the underlying dynamics produce. Can they probably compare and discuss how the performance of such approaches compares to the performance of the proposed framework?

- What does "empirically desirable dynamics" mean (line 179 second column).

- In practice, how should one go forward and chose one of the proposed gauge regularizers to employ?

**Limitations:**

- Yes

**Strengths And Weaknesses:**

## Strengths

- I like the main take home message of the paper, that restricting the dynamics to gradient fields is not inherent to the population-dynamics inference problem itself, but corresponds to one particular gauge choice. This is relevant for most population dynamics inference methods. ( **originality** + **significance** )
- I find the weak continuity formulation approach insightful and elegant. (**soundness** + **originality** )
- The tracer example the authors provide is a convincing example that different gauges might correspond to different learned dynamics. (**soundness** + **presentation** )

----


## Weaknesses

- The authors provide somewhat narrow experimental validation. (**soundness**)

- The presentation could be improved in terms of notation. Since non-gradients fields are by default multi-dimensional, the authors do not employ proper boldfacing for vectors in their equations. (**presentation**)

- The authors have not actually implemented the competing methods, whose results are presented in table 1, but rather use the results mentioned in Blickhan et al 2025. While I understand the incentive, this might sometimes lead to comparing frameworks that have been applied to slight different settings/data if one does not directly control inference themselves. (**soundness**)



### Minor

- Line 280 : "pre-compute**d**"
- Line 304: "the" seems extra

---

> ### Author Rebuttal · Authors · 2026-03-30
>
> We thank the reviewer for their careful reading. We address each point below.
>
> ---
>
> ***"The authors have not actually implemented the competing methods... might sometimes lead to comparing frameworks that have been applied to slightly different settings/data"***
>
> To address the reviewer's comment, we have run our implementation of DICE, HOAM, and AM on the tracer particle example (Figure 2) and report the total variation distance (see Figure 3) in the table:
>
> | | NGIF-div  | NGIF-curl | NGIF-kin  | grad-flow | DICE | HOAM | AM |
> |-|-|-|-|-|-|-|-|
> | final T.V.D | 0.51 | 0.48 | 0.49 | 0.62 | 0.61|  0.57 | 0.72 |
>
> Additionally we run HOAM, DICE, and AM on our high dimensional turbulence and compute the relative error in the enstrophy over time:
>
> | | NGIF | DICE | HOAM | AM   |
> |-|-|-|-|-|
> | avg enstrophy err | 0.09 | 0.71 | 0.67 | 0.69 |
>
> Both of these new results agree with our NGIF-based grad-flow method, providing further evidence that constraining dynamics to a gradient parameterization can be overly restrictive and is hard to train (as discussed in Appendix C of Neklyudov et al).
>
> ***"The authors provide somewhat narrow experimental validation."***
>
> We added 3 new studies: (1) a $\lambda$-robustness sweep showing marginal-matching accuracy is stable across orders of magnitude of $\lambda$ (see the response below to reviewer DoVi); (2) head-to-head comparisons against DICE, HOAM, AM on the tracer-particle and high-dim. turbulence benchmarks; and (3) a dimensionality scaling study up to $d=64$ and a non-latent $64 \times 64$ PDE (see reviewer tFXm). These address robustness, controlled baselines, and scalability respectively.
>
> ***"Can the authors better justify the trade-off formulation versus the constrained formulation?"***
>
> We appreciate this question. As the reviewer identifies, Section 5 discusses the alternative of minimizing the gauge regularizer subject to the weak continuity constraint (4). In practice, enforcing hard constraints over neural-network-parameterized velocity fields requires iterative constrained optimization machinery (e.g. augmented Lagrangian methods). Hard-enforcing an approximate constraint adds substantial optimization complexity.
>
> The penalized form (Eq. 6) yields a single unconstrained objective amenable to standard SGD methods with no inner-outer loop. It is simple to implement, stable in practice, and as our experiments show, effective at enforcing marginal matching across all benchmarks.
>
> ***"In practice, how should one go forward and choose one of the proposed gauge regularizers to employ?"***
>
> An important point is that there is no single "correct" regularizer in the sense of marginal-matching accuracy. All 3 regularizers we consider yield good distributional results, because it is the weak continuity-equation loss, not the regularizer, that enforces agreement with the observed time marginals.
>
> There are reasons to prefer one regularizer over another when trajectory-level structure matters. The regularizers encode transparent physical priors. For examples, $\mathcal{G}_{\mathrm{DIV}}$ yields trajectories closest to the physical ones in the incompressible flow setting.
>
> We view opening this design space as a key contribution: the field has largely treated the minimal-kinetic-energy gauge as the only option, and our framework makes gauge selection an explicit, modular choice. The three regularizers we propose are not exhaustive; practitioners can design problem-specific ones as needed, much as one designs priors in other settings.
>
> ***"How sensitive is the method to the exact choice of the test function family?"***
>
> Any family that is dense in the admissible test space $\Phi$ (as stated in Section 2.2) is valid. Random Fourier features are a particularly natural choice: they span the required space, and all needed quantities ($\phi$, $\nabla\phi$, $\Delta\phi$) are available in closed form at $\mathcal{O}(d)$ cost.
>
> ***"The authors only briefly mention...compare and discuss how the performance of such approaches compares?"***
>
> These methods (Solomon, 2023; Kapuńiak et al., 2024) operate in the two-marginal setting and curve interpolation paths via a learned metric, whereas we learn a velocity field consistent with an entire sequence of marginals. A direct comparison would require non-trivial adaptation. Moreover, both still implicitly fix the gauge: the former uses OT paths under the learned metric (i.e. they are still the minimal kinetic energy ones), the latter determines paths implicitly via the choice of stochastic interpolant. We will add these citations and discussion in Section 4.
>
> ***"What does 'empirically desirable dynamics' mean?"***
>
> We mean trajectories that align with prior physical knowledge about the system. For example, using $\mathcal{G}_{\mathrm{DIV}}$ for an incompressible fluid setting. We will replace be more precise in the revision.
>
> ***"boldfacing for vectors...", typos***
>
> We will adopt bold notation for vector quantities. Typos will be corrected.

---

> > ### Author Rebuttal · Reviewer_vAsx · 2026-04-04
> >
> > I would like to thank the authors for engaging with my and the other reviewers' reviews. My concerns were partially resolved, but I still have some follow up questions.
> >
> > **My main remaining concern is how, in practice, one would determine which of the well-performing admissible velocity fields is actually closest to the ground-truth field, and whether the authors can provide some recipe/heuristic on this.**
> >
> > Since the table you proved (first one for particle tracer experiment) corresponds to Figure 3 where you report the evolution of the distances, would it be possible to provide a link with a Figure with the additional implemented frameworks plotted along with appears in Figure 3 of the submission?
> >
> > >An important point is that there is no single "correct" regularizer in the sense of marginal-matching accuracy. All 3 regularizers we consider yield good distributional results, because it is the weak continuity-equation loss, not the regularizer, that enforces agreement with the observed time marginals.
> > >There are reasons to prefer one regularizer over another when trajectory-level structure matters. The regularizers encode transparent physical priors. For examples,  yields trajectories closest to the physical ones in the incompressible flow setting.
> > >We view opening this design space as a key contribution: the field has largely treated the minimal-kinetic-energy gauge as the only option, and our framework makes gauge selection an explicit, modular choice. The three regularizers we propose are not exhaustive; practitioners can design problem-specific ones as needed, much as one designs priors in other settings.
> >
> > I think the authors should discuss in the revised manuscript how one would choose the regularizer in practice. In particular, they should acknowledge that this choice is likely to be strongly system-dependent, and that establishing principled selection criteria remains an open question for future work.
> >
> >
> > I would like to thank the authors again.

---

> > > ### Author Response · Authors · 2026-04-06
> > >
> > > ***would it be possible to provide a link with a Figure with the additional implemented frameworks plotted along with appears in Figure 3 of the submission?***
> > >
> > > No problem. We provide the additional plots corresponding to our new results here: https://anonymous.4open.science/r/ngif_rebuttal-4624/res.pdf. We will integrate these into our revision.
> > >
> > > ***I think the authors should discuss in the revised manuscript how one would choose the regularizer in practice. In particular, they should acknowledge that this choice is likely to be strongly system-dependent, and that establishing principled selection criteria remains an open question for future work.***
> > >
> > > We agree. This is an important question and a new discussion will feature substantially in our revision. We can give a briefly outline of what we would like to to say and what a principled selection criteria might look like:
> > >
> > > 1) Choosing a regularizer may depend on some explicit physical prior or desiderata for trajectory-level behaviour. Div, Curl, Kin are all examples of what this might look like in practice; but the whole space of what is possible is much broader than what we can explicitly consider in this paper. In the case of Div for the tracer particles we believe this a good example of what a principled choice might be as the underlying vector field is known to be divergence-free.
> > >
> > > 2) Choosing a regularizer can also be done if one has explicit data outside the standard time-marginal samples. Our framework can incorporate it directly as a gauge regularizer. For example, suppose one knows the true velocity field is not exactly divergence-free but has a known spatiotemporal divergence profile $d(x,t)$. One can define a modified regularizer $\mathcal{G}(v) = \|\nabla \cdot v(x,t) - d(x,t)\|^2$ and use it as the gauge penalty. Indeed, our $\mathcal{G}_{\mathrm{DIV}}$ tracer experiment in Section 3.2 is simply the case where $d(x,t)$ is uniformly zero.
> > >
> > > We thank the reviewer again for their detailed comments and discussion which has certainly improved the quality of the paper.

---

### Decision · Program_Chairs · 2026-04-30

**Decision:**

Accept (regular)

**Comment:**

This paper proposes Non-Gradient Inference Flows (NGIF), a framework for learning population dynamics from time-marginal snapshots that decouples marginal matching from velocity-field selection, exposing gauge freedom that existing methods implicitly resolve by hard-coding a minimal-kinetic-energy (gradient) prior. The paper received uniformly positive reviews from confident reviewers, all of whom recognized the novelty and elegance of reframing the gradient-field assumption as one particular gauge choice rather than an inherent constraint. The weak continuity formulation using random Fourier features is insightful, and the tracer particle example effectively demonstrated that different gauge regularizers yield meaningfully different learned dynamics while preserving distributional accuracy. I recommend for the acceptance for this paper.